# In Vitro Prebiotic Effects and Antibacterial Activity of Five Leguminous Honeys

**DOI:** 10.3390/foods12183338

**Published:** 2023-09-06

**Authors:** Florinda Fratianni, Beatrice De Giulio, Antonio d’Acierno, Giuseppe Amato, Vincenzo De Feo, Raffaele Coppola, Filomena Nazzaro

**Affiliations:** 1Institute of Food Science, CNR-ISA, Via Roma 64, 83100 Avellino, Italy; fratianni@isa.cnr.it (F.F.); bdegiulio@isa.cnr.it (B.D.G.); dacierno.a@isa.cnr.it (A.d.); defeo@unisa.it (V.D.F.); coppola@unimol.it (R.C.); 2Department of Pharmacy, University of Salerno, Via Giovanni Paolo II, 132, 84084 Fisciano, Italy; 3Department of Agriculture, Environmental and Food Sciences, University of Molise, Via de Sanctis, 86100 Campobasso, Italy

**Keywords:** honey, prebiotics, probiotics, antioxidant, biofilm

## Abstract

Honey is a natural remedy for various health conditions. It exhibits a prebiotic effect on the gut microbiome, including lactobacilli, essential for maintaining gut health and regulating the im-mune system. In addition, monofloral honey can show peculiar therapeutic properties. We in-vestigated some legumes honey’s prebiotic properties and potential antimicrobial action against different pathogens. We assessed the prebiotic potentiality of honey by evaluating the antioxidant activity, the growth, and the in vitro adhesion of *Lacticaseibacillus casei*, *Lactobacillus gasseri*, *Lacticaseibacillus paracasei* subsp. *paracasei*, *Lactiplantibacillus plantarum*, and *Lacticaseibacillus rhamnosus* intact cells. We also tested the honey’s capacity to inhibit or limit the biofilm produced by five pathogenic strains. Finally, we assessed the anti-biofilm activity of the growth medium of probiotics cultured with honey as an energy source. Most probiotics increased their growth or the in vitro adhesion ability to 84.13% and 48.67%, respectively. Overall, alfalfa honey best influenced the probiotic strains’ growth and in vitro adhesion properties. Their radical-scavenging activity arrived at 83.7%. All types of honey increased the antioxidant activity of the probiotic cells, except for the less sensitive *L. plantarum*. Except for a few cases, we observed a bio-film-inhibitory action of all legumes’ honey, with percentages up to 81.71%. Carob honey was the most effective in inhibiting the biofilm of *Escherichia coli*, *Listeria monocytogenes*, *Pseudomonas aeruginosa*, and *Staphylococcus aureus*; it retained almost entirely the ability to act against the bio-film of *E. coli*, *L. monocytogenes*, and *S. aureus* also when added to the bacterial growth medium instead of glucose. On the other hand, alfalfa and astragalus honey exhibited greater efficacy in acting against the biofilm of *Acinetobacter baumannii*. Indigo honey, whose biofilm-inhibitory action was fragile per se, was very effective when we added it to the culture broth of *L. casei*, whose supernatant exhibited an anti-biofilm activity against all the pathogenic strains tested. Conclusions: the five kinds of honey in different ways can improve some prebiotic properties and have an inhibitory biofilm effect when consumed.

## 1. Introduction

Honey has been considered a natural remedy for various health conditions for centuries. It is a natural source of carbohydrates, which can provide a quick energy boost. Its content of vitamins and minerals can also help in improving overall health [1]. Honey is an effective cough suppressant, especially in children, acts as a cardio protective, liver-protective, and anti-diabetic agent, and exhibits immunomodulatory, antitumor, neuroprotective, cytotoxic, antithrombotic, hypocholesterolemic activities [2]. Clinical investigations have also demonstrated the effectiveness of honey in treating medical conditions such as diarrhea, gastritis, gastric-duodenal ulcers, dermatitis, arthritis, and as a wound healing agent [3,4]. With its antioxidant and anti-inflammatory properties, it can help mitigate some pain, such as sore throat, and through coating the inflamed tissue, thus reducing the irritation. In addition, the antioxidant activity of honey can help boost the immune system and protect the body against diseases. Honey contains various sugars, such as fructose and glucose, which are not fully digested in the small intestine. Therefore, it exhibits a prebiotic effect on the gut microbiome, mainly for certain beneficial bacteria, such as bifidobacteria and lactobacilli, essential for maintaining gut health and regulating the immune system. Research has shown that regular and not excessive consumption of honey can help reduce inflammation in the gut. Honey can positively affect overall health, as the gut microbiome plays a crucial role in many aspects of health, including digestion, immune function, and mental health [5]. Honey is known for its antimicrobial properties. Moreover, its effectiveness against a wide range of microorganisms is documented in scientific research. The antimicrobial activity of honey is multifactorial, mainly due to the presence and content of different molecules, such as polyphenols (phenolic acids and flavonoids), peroxides, and sugars, and low water activity, which in different manners lead to important antibacterial activity. The high sugar content creates a hypertonic environment that draws water out of bacterial cells, causing them to shrink and die. The low water activity determines an unsuitable environment for the growth of several pathogenic bacteria. With their antimicrobial properties, polyphenols, such as phenolic acids and flavonoids, inhibit the growth of various pathogenic microorganisms, including bacteria such as *Staphylococcus aureus*, *Escherichia coli*, and *Salmonella* spp., as well as fungi such as *Candida albicans*. Honey’s antibacterial properties can help heal wounds, and prevent infection. Honey can limit the formation or the growth of biofilms, complex communities of microorganisms, including bacteria, growing on surfaces and are embedded in a self-produced matrix of extracellular polymeric substances [6,7]. Bacterial biofilms are often associated with chronic infections and can significantly affect human health. They can give rise to an increase in bacterial virulence, referring to their ability to cause disease. Biofilm matrix can protect bacteria from host immune defenses and antibiotics, making it difficult to eradicate the infection. However, it can also act as a reservoir for bacteria, allowing them to persist and continuously release virulence factors. Furthermore, biofilms can facilitate the transfer of genetic material between bacteria, leading to the acquisition of new virulence traits. Honey produced by bees that collect most of the nectar from a single flower species is the so-called monofloral honey. In contrast, the so-called multi-floral honey originates from the nectar of many different flowers. Monofloral and multifloral honeys have different flavors, chemical ingredients, and physicochemical and therapeutic properties [8]. Bees that primarily collect nectar from leguminous plants produce legume kinds of honey. Some of the most popular types of leguminous honey include clover, acacia, and alfalfa honey. Leguminous honey is often used as a natural sweetener and an ingredient in baking and cooking. They also have several health benefits, including antibacterial and antioxidant properties. Most studies on monofloral honey from leguminous flowers refer to well-known honey, such as acacia or clover honey. However, the biodiversity that legumes can offer is enormous. Therefore, the heritage that legumes can represent, also from the point of view of the products deriving from them, such as honey, can further increase the added value even of neglected leguminous species. In Italy, around 100,000 hectares are planted with legumes, producing 190,000 tons (https://blog.wetipico.it/i-legumi-italiani-alla-scoperta-della-loro-importanza/ last access 2 July 2023). In particular, there is an increase in the cultivation of chickpeas, lentils, and peas. Italian production differs in quality and cultural importance. The resumption of legume cultivation, particularly in the central and southern regions, has developed small productions of ancient varieties. Thus, throughout the national territory, species of native legumes previously abandoned are being rediscovered. Therefore, even if the numbers are still small, cultivating these plants is fundamental for the plant biodiversity of the country. Our work aimed to investigate the prebiotic properties of five leguminous honeys, alfalfa, astragalus, carob, indigo, and sainfoin, and their potential antimicrobial action against different pathogens. We evaluated their prebiotic potentiality using *Lacticaseibacillus casei*, *Lactobacillus gasseri*, *Lacticaseibacillus paracasei* subsp. *paracasei*, *Lactiplantibacillus plantarum*, and *Lacticaseibacillus rhamnosus* as probiotic tester strains, assessing the capacity of the honey to affect their growth and in vitro adhesive capacity, as well as the antioxidant ability exhibited by their intact cells, which are parameters affecting their prebiotic properties [9,10]. We also tested the capacity of the alfalfa, astragalus, carob, indigo, and sainfoin’ honey to inhibit or limit the biofilm produced by five pathogenic strains: *Acinetobacter baumannii*, *Escherichia coli*, *Listeria monocytogenes*, *Pseudomonas aeruginosa*, and *Staphylococcus aureus*. Finally, we assessed the antibiofilm activity of these honeys, when used in the growth medium of the probiotics as a substitute for glucose, against the pathogens mentioned above.

## 2. Materials and Methods

### 2.1. Honey

Five organic Italian commercial monofloral honey: alfalfa (*Medicago sativa* L., from Tuscany), astragalus (*Astragalus nebrodensis (Guss.) Strobl.*, from Sicily), carob (*Ceratonia silique* L., from Sicily), indigo (*Indigofera tinctoria* L., from Lumbardy), and sainfoin (*Onobrychis viciifolia* Scop., from Abruzzo), were used for our experiments (Figure 1). The monofloral character was indicated by the companies and respected the Italian law 179 of 2004, which also legislates on the floral or vegetable origin, if the product is wholly or mainly obtained from the indicated plant and possesses its organoleptic, physicochemical and microscopic characteristics. Three packages of each type of honey were used, making sure that they belong to the same year of production. None of the samples showed crystallization at the time of purchase. The transportation was performed in polystyrene bags with little ice-bags, so to maintain as most as possible the original characteristics of honey, which was then stored at room temperature (20 °C) in the dark until the analyses; later, we removed an aliquot from each of the three samples of each honey, which was strongly mixed with sterile phosphate-buffered saline (PBS) (rate: 1 g of honey in 4 mL of PBS), filtered (0.45 µm, Millipore, Milano, Italy) and subjected to the microbial tests.

### 2.2. Prebiotic Activity of the Honey

Five strains of lactic acid bacteria (LAB), *Lacticaseibacillus casei* Shirota (LcS), *Lactobacillus gasseri* LG050, *Lacticaseibacillus paracasei* subsp. *paracasei* I 1688, *Lactiplantibacillus plantarum* 299V and *Lacticaseibacillus rhamnosus* GG were obtained from commercial formulation available in a local pharmacy. The choice to use commercial strains originates from the fact that their probiotic properties have already been ascertained, even at a clinical level, compulsory for their commercialization as probiotics.

#### 2.2.1. Growth of Lactic Acid Bacteria in the Presence of Honey

The strains were grown at 37 °C (except *L. plantarum*, grown at 30 °C) for 16–18 h in MRS without glucose (Liofilchem, Roseto degli Abruzzi, Italy), which was substituted by an equal concentration (*weight*/*vol*) of the five honeys. The growth was read at λ = 600 nm (Cary 50Bio, Varian, Palo Alto, CA, USA). The effect of the five honeys on the growth of the lactic bacteria was calculated as a percentage with respect to the control when the strains were grown in the presence of glucose.

#### 2.2.2. Microbial Adhesion to Solvent

The microbial adhesion to solvent (MAS) test was performed according to Nazzaro et al. [11] with some modifications. First, LAB cells were washed with sterilized isotonic saline (0.9%), harvested, and re-suspended in the same solution so that the final concentration of intact cells was the same as in the initial experiment. The absorbance of the cell suspension (A0) was measured at λ = 600 nm (Cary50Bio Varian); then, we added an equal volume of xylene and mixed thoroughly the two-phase system by continuously vortexing for 3 min. The aqueous phase was removed after one h of incubation at 37 or 30 °C (depending on the strain), and its absorbance (A1) was measured. The adhesion was calculated from three replicates as a percentage decrease in the optical density of the original bacterial suspension by the formula: % = [(A0 − A1)/A0]·100

#### 2.2.3. Antioxidative Activity of Lactic Acid Bacteria Grown in the Presence of the Five Legumes’ Honeys

LAB were grown in an MRS medium, where glucose was substituted by an equal concentration (*w/vol*) of each honey. Except for *L. plantarum*, grown at 30 °C, the strains were grown at 37 °C for 18 h. Then, aliquots of the culture were transferred to 15 mL tubes, centrifuged (3000 rpm, 4 °C, 10 min), and washed with sterilized isotonic saline (0.9%) three times. The final concentration of intact cells was the same as in the initial experiment.

##### Reducing Power Capacity

The reducing power capacity of intact cells of LAB was evaluated following the method of Lin et al. [12]. Briefly, 0.5 mL of LAB were mixed with 0.5 mL of phosphate buffer 0.02 M pH 6.6 and 0.5 mL of 1% potassium ferricyanide. The mixture was incubated at 50 °C in a water bath for 20 min. After cooling, 0.5 mL of 10% trichloroacetic acid was added to the mixture, which was then centrifuged at 3000 rpm for 10 min. One ml of the upper phase was mixed with 1.0 mL of FeCl_3_ 0.1%. The absorbance was measured at λ = 700 nm (Cary Bio Varian, Palo Alto, CA, USA). A blank was prepared without adding honey. Cysteine at various concentrations (from 0.01 to 10 mM) was used as standard for the expression of the reducing activity. As known, Fe^3+^ is transformed to Fe^2+^ in the presence of possible reducing power. The increase in absorbance of the reaction mixture indicates an increase in reducing power.

##### The Anti-Lipid Peroxidation Activity (ILAP, Inhibition of Linoleic Acid Peroxidation)

The measurement of the antioxidative activity of intact LAB cells was performed by the thiobarbituric acid (TBA) method, based on the monitoring of inhibition of linoleic acid peroxidation [13]. A Fe/H_2_O_2_ system was used for the catalysis of oxidation. A phosphate buffer solution (0.5 mL, 0.2 M, pH 7.4), 0.5 mL of linoleic acid emulsion, 0.2 mL of FeSO_4_ 0.01%, 0.2 mL of H_2_O_2_, and 0.5 mL of intact cells were mixed and incubated at 37 °C. Blank samples contained deionized water. After 12 h of incubation, 2 mL of the reaction solution was mixed with 0.2 mL of trichloroacetic acid TCA) 4%, 2 mL of TBA (0.8%), and 0.2 mL of butylated hydroxytoluene (BHT) (0.4%) to stop further sample peroxidation while processing. This mixture was incubated at 100 °C for 30 min and cooled. After centrifugation (t12,000 rpm for 5 min), the absorbance was measured at λ = 532 nm (Cary50Bio, Varian, Palo Alto, CA, USA). 

The percentage of inhibition of linoleic acid peroxidation was defined as follows: % = [(A_532sample_)/(A_532blank_)]·100

##### Hydroxyl Radical Scavenging Activity

The hydroxyl radicals scavenging activity of LAB grown in the presence of the honeys were determined following the method of Guo et al. [14]. The hydroxyl radicals were produced by the Fenton reaction occurring between H_2_O_2_ and FeSO_4_. The reaction was performed in 1.0 mL of 5 mM sodium salicylate, 1.0 mL of 5 mM FeSO_4_, 1.0 mL of LAB, and 1.0 mL of 3 mM H_2_O_2_. The reaction mixture was incubated for 1 h at 37 °C. The absorbance was measured at λ = 510 nm (Cary50Bio, Varian, Palo Alto, CA, USA). Distilled water was used as a control. The percent scavenging rate was calculated following the formula:


% = [1 − (A_510sample_/A_510control_)]·100


##### Scavenging Activity of LAB Strains

The scavenging effect of the LAB strains grown in the presence of the honeys on the free radical α,α-diphenyl-β-picrylhydrazyl (DPPH) was measured in accordance with the original protocol of Singleton and Rossi [15], but slightly modified to be adapted to the use of intact cells of LAB as possible antioxidants. One ml of LAB culture, previously washed with sterile physiological solution, and 1 mL and a freshly prepared DPPH solution (6 × 10^−5^ M, 1 mL, Sigma—Aldrich, St. Louis, MO, USA) were mixed. The mixture was vigorously shaken and left to react for 1 h in the dark at room temperature. The control contained water instead of the sample solution. The scavenged DPPH was then monitored by determining the absorbance at λ = 517 nm (Cary_50_Bio, Varian).

### 2.3. Antibiofilm Activity Exhibited by the Legumes Honey

#### 2.3.1. Microorganisms and Culture Conditions

The bacterial strains *Acinetobacter baumannii* (ATCC 19606), *Escherichia coli* (DSM 8579), *Pseudomonas aeruginosa* (DSM 50071), *Listeria monocytogenes* (ATCC 7644), and *Staphylococcus aureus* subsp. *aureus* Rosebach (ATCC 25923), purchased from Deutsche Sammlung von Mikroorganismen und Zellkulturen (DSM, Braunschweig, Germany), were used in the experiments. Before the antimicrobial assays, they were cultured in Luria Broth for 18 h at 37 °C (*A. baumannii* was grown at 35 °C) and 80 rpm (Corning LSE, Pisa, Italy).

#### 2.3.2. Minimal Inhibitory Concentration (MIC)

The resazurin microtiter-plate assay evaluated the MIC [16,17]. The tests were performed in flat-bottomed 96-well microtiter plates, which were incubated at 37 °C for 24 h (*A. baumannii*, grew at 35 °C under the same conditions). The MIC value was revealed by the visual color change from dark purple to colorless. Sterile DMSO and tetracycline (dissolved in DMSO, 1 mg/mL) were used as negative and positive control, respectively. Determinations were performed in triplicate; the results were expressed as the arithmetic mean ± standard deviation.

#### 2.3.3. Inhibition of Biofilm Formation

The capacity of the honeys to affect the bacterial biofilm formation was assessed in flat-bottomed 96-well microtiter plates (Falcon, VWR International, Milano, Italy) [17]. The overnight bacterial cultures were adjusted to 0.5 McFarland with fresh culture broth. Later, 10 µL of the bacterial cultures, 10 µg/mL or 20 µg/mL of honey and sterile Luria–Bertani broth (LB, Sigma Aldrich Italia, Milano, Italy) were brought in each well to reach a final volume of 250 µL. Microtiter plates were covered with parafilm tape to preclude the evaporation of material included in the wells and incubated for 48 h at 37 °C (35 °C for *A. baumannii*). Planktonic cells were removed, then the attached cells were lightly washed twice with sterile phosphate buffered saline (PBS), which was discarded. The plates were kept for 10 min under a flow laminar hood before the addition of 200 µL of methanol in each well for 15 min to allow the fixation of the sessile cells. Methanol was discarded, and each plate was left to let the dryness; then 200 µL of 2% *w*/*v* crystal violet solution were added to each well. After 20 min, the staining solution was removed, and the plates were lightly washed with sterile PBS and left to dry. The addition of 200 µL of glacial acetic acid 20% *w*/*v* let the release of the bound dye. The absorbance was measured at λ = 540 nm (Cary50Bio, Varian). The percent value of adhesion was calculated with respect to the control (represented by the bacterial cells grown without the presence of the samples, which inhibition rate was assumed as 0%). Triplicate tests were performed, and results were expressed as the mean ± SD.

### 2.4. Antibacterial and Antibiofilm Activity of the Supernatants of the LAB Grown in the Presence of the Honeys

The five LAB strains were grown in MRS medium, in which glucose was substituted by an equal concentration (*w*/*vol*) of the honey. Except for *L. plantarum*, grown at 30 °C, the strains were grown at 37 °C for 18 h. Then, the tubes were centrifuged (3000 rpm, 4 °C, 10 min). The supernatant was recovered and filtered (mesh 0.22 μm, Merck Life Science, Milano, Italy) to carry out the antibacterial and antibiofilm tests.

For the antibacterial activity test, five pathogenic strains were grown in Luria Bertani broth at 37 °C (*A. baumannii* was incubated at 35 °C) for 18 h. After the evaluation of the bacterial growth, conducted by reading at λ = 600 nm, 10 μL of each bacterial culture were added to a multiwell, previously filled with 40 μL/mL and 80 μL/mL of LAB culture supernatant and Luria-Bertani (Merck Life Science, Milano, Italy) broth up to a final volume of 250 μL. After 24 h of incubation at 37 °C (35 °C for *A. baumannii*), the growth of the pathogenic microorganisms was evaluated, comparing the inhibition (calculated in percentage) with respect to untreated bacteria (for which the inhibition was rated = 0%).

For the biofilm inhibition test performed in the presence of the LAB supernatant grown in the presence of glucose (control) and an equal quantity of honey, 40 μL/mL and 80 μL/mL of LAB culture supernatant were added to the pathogenic cultures. The multiwell plates were incubated at 37 °C or 35 °C for 48 h. The inhibitory effect on the adhesion process of pathogens was evaluated following the protocol of Fratianni et al. [17], using the previously described crystal violet, and it was measured as percent respect to the control (untreated pathogenic bacteria) for which it was assumed an inhibition = 0%.

### 2.5. Statistical Analysis

Data were expressed as the mean ± SD of three experiments and statistically analyzed using a two-way ANOVA followed by Dunnett’s multiple comparison test, at the significance level of *p* < 0.05, using GraphPad Prism 6.0 (GraphPad Software, Inc., San Diego, CA, USA).

## 3. Results and Discussion

### 3.1. Influence of the Legume Honeys on the Growth and In Vitro Adhesive Capacities of Probiotic Bacteria

Several investigations reported the potential prebiotic effect of honey, investigating their impact on probiotics commonly present in the human intestinal system. Most of them are referred to the honey’s capacity to stimulate the growth of beneficial bacteria [18,19]. Others are focused on the ability of honey to influence the hydrophobicity characteristics, the self-aggregation [20,21], and to enhance the hypocholesterolemic effects of lactic acid bacteria [22]. Moreover, others reported the honey’s impact on lactobacilli’s probiotic activity, such as the effect on lactic acid bacteria [23,24]. In our work, the impact that the five leguminous honeys could exert on some biological characteristics of probiotic bacteria have been investigated. In particular, the effect of honey used instead of glucose in the MRS culture broth on the growth and in vitro adhesion of lactobacilli, and their antioxidant activity have been studied. Possible differences between the antibiofilm activity of LAB growth supernatants after appropriate filtration and control have been investigated. In this case, the honeys were tested at the same presumptive concentrations as those which saw it as an ingredient in the culture broth. 

The influence of honeys on growth and in vitro adhesive capacity of the five strains of probiotic bacteria has been analyzed, in particular, by verifying the difference with their growth under conventional conditions (in MRS broth). For this purpose, the honeys have been added to a glucose-free MRS broth at the same concentration as that of glucose present in the conventional MRS broth used as a control, and the influence of the honey on the bacterial growth was evaluated (in terms of percentage with respect to the control. The results are shown in the Figure 2 (for the comparison of the bacterial growth) and Figure 3 (for the comparison of the in vitro adhesive capacity of the bacterial cells).

The analysis of data on the growth of the probiotic strains (Figure 2) highlighted several interesting results. The presence of astragalus honey, with the sole exception of *L. plantarum*, determined an increase in bacterial growth, which exceeded that of the control by 44.73% (*L. casei* Shirota) and by 39.27% (*L. gasseri*). However, the influence of this honey on bacterial adhesion (Figure 3) was different. The probiotic strains that exhibited the best performance did not improve their adherence to control. Conversely, *L. plantarum*, *L. rhamnosus*, and *L. paracasei*, for which the presence of astragalus honey had a slight effect on bacterial moltiplication, exhibited a significant increase in their in vitro adhesion capacity, which ranged between 17.01% (*L. plantarum*) and 30.74% (*L. paracasei*) more than the control.

The presence of carob honey in the probiotic culture broth determined a more significant effect on the growth of *L. gasseri* and *L. casei* Shirota. Concurrently, *L. rhamnosus* exhibited a similar increase of percentage both as regards the growth (13.64%) and in terms of adhesion capacity (16.15%) compared to control. The presence of carob honey did not determine any positive effect on the growth of *L. paracasei* compared to the control. However, this honey affected its adhesion capacity, which exceeded that of the corresponding control strain by 22.78%.

Alfalfa honey determined an increase in growth almost double when added to the culture broth of *L. gasseri* (87.21% increase and *L. casei* Shirota (71.64% increase) compared to the respective controls, confirming to be an excellent growth-stimulating, at least for these two LAB strains [25,26]. Also in this case, when it did not influence the growth, however it had a massive effect on the LAB in vitro adhesion capacity, as in the case of *L. paracasei*, which adhesion increased up to 44.71% compared to the control. A similar behaviour was observed when indigo honey instead of glucose was used, with a consistent increase in the growth of *L. gasseri* (84.13% more than the control) and of *L. casei* Shirota (76.12% more than the control), and to the adhesion of *L. paracasei*, with a value of 34.14% more than control. Sainfoin honey equally affected the growth of *L. gasseri* (66.43% more than the control) and *L. casei* Shirota (79.99% more than the control), while it was able to increase the in vitro adhesion capacity of *L. paracasei* (48.67% more than control). Therefore, the honeys positively affected the growth of *L. gasseri* and *L. casei* Shirota. At the same time, in the case of *L. paracasei*, we observed a positive action, especially on the adhesion capacity of this probiotic, regardless of the honey used. The adhesive capability of *L. rhamnosus* was mainly influenced by astragalus honey. Carob honey exerted a beneficial effect on both the growth and adhesion capabilities of this probiotic strain. The other honeys (indigo, sainfoin, and alfalfa) did not affect its growth or adhesive properties. No honey, on the other hand, influenced positively the growth of *L. plantarum*. However, astragalus honey positively influenced its in vitro adhesion (17% increase). This honey was very effective in stimulating the growth of *L. gasseri* and *L. casei* Shirota. On the other hand, the effect on *L. plantarum* was practically nil, in accordance with what observed by Rakabizadeh and Tadayoni [27]. To our knowledge, it is the first time in which astragalus honey has been studied to evaluate its prebiotic properties on different LAB strains. However, the prebiotic properties of some metabolites of the astragalus genus are known, such as the polysaccharides, which can increase the population of *L. gasseri* and *L. amylovorus* in weaned piglets [28], and that, when fermented, help to implement calcium absorption, hindering osteoporosis [29]. The extract of astragalus affected the growth of *L. casei*, thus obtaining a fermented product with anti-inflammatory properties. The extract of an *Astragalus membranaceus* Moench extract has been shown to influence the growth of *L. casei*, thus obtaining a fermented product with anti-inflammatory properties [30]. The growth-stimulating effect exhibited by carob honey on *L. gasseri*, *L. rhamnosus*, and *L. casei* Shirota, never previously reported, confirmed the positive effect that carob could exert on the growth of different lactic acid bacteria [31,32], although, in our case, the experiment was performed using carob honey, not a carob extract. The growth-stimulating effect of alfalfa honey, which resulted particularly efficient on *L. gasseri* and *L. casei* Shirota, confirms previous studies in which this honey increased the count of *Streptococcus*, lactobacilli, and *Bifidobacterium* strains [33]. It is significant that, as far as growth is concerned, *L. casei* Shirota has shown a different behavior from that of *L. paracasei* and *L. rhamnosus*, generally closely phenotypically and genotypically related, belonging to the L. casei group [34]. Its behavior resulted somewhat similar to that of *L. gasseri*, which instead is taxonomically linked to the group of *L. acidophilus* [35]. To our knowledge, there are no studies on the prebiotic effect of indigo nor of the sainfoin honeys. However, secondary metabolites of indigo could positively affect the gut microbiome, alleviating symptoms of inflammatory bowel disease [36] or by influencing senescence processes [37]. On the other hand, an extract of sainfoin exhibited a general positive influence on gut microbial communities [38], and resulted helpful also for feeding, for example as an anti-helmintic [39].

### 3.2. Influence of the Legumes’ Honey on the Antioxidant Capacity of Probiotic Cells

Reactive oxygen substances (ROS) can provoke several damages to cells, promoting, when in excessive amounts, chronic inflammatory, dysmetabolic, cardiovascular, and neurodegenerative diseases, and cancer [40,41]. Our body synthesizes antioxidant enzymes and molecules that, together with the food antioxidants, build a biological antioxidant barrier. Nevertheless, in some conditions, the defense system is insufficient, so the possibility of increasing antioxidant defenses becomes essential to maintain human health and disease prevention [42]. In this direction, an interesting approach can be the study of probiotics’ antioxidant activity, such as that exhibited by *L. rhamnosus*, and their counteracting action against the oxidative stress in the host, thus contributing to decreasing the risk of accumulation of ROS [43]. Several studies have shown that many probiotics can scavenge hydroxyl radicals and superoxide anions in vitro and in vivo, also improving oxidative stress in patients with type 2 diabetes [44,45]. LAB can generally resist ROS, and a diet including probiotics can also protect normal liver functions [46,47]. Several studies, such as that of Amaretti et al. [9], ascertained such properties for some lactobacilli, including *L. plantarum*. More recently, Won et al. demonstrated the antioxidant activity of *L. plantarum* and *L. paracasei* [48]. Bee honey intrigued the scientific community’s interest as a natural dietary antioxidant [49]. However, the scientific literature lacks data regarding the effect that honeys, particularly from leguminous plants, can exert on the antioxidant activity of probiotic cells. Starting from these considerations, we evaluated the potential effect that leguminous honeys could exert, when added to the growth medium in place of glucose, on some probiotic strains’ antioxidant capacity through the use of different tests. Results are shown in Table 1 (panels a–d). 

Data of two radical scavenging assays (HRS and DPPH) showed that the intact LAB cells might generally inhibit the formation of the two radicals. When the scavenging activity was evaluated through the DPPH method, the results highlighted a different behavior of the probiotic cells, depending on the type of honey used instead of glucose in the constitution of the MRS culture broth (Table 1a). Astragalus and sainfoin honeys were the most effective in increasing the antioxidant capacity of probiotic bacteria. Astragalus honey increased the antioxidant performances of all probiotics (except *L. casei* Shirota). In some cases, for example for *L. gasseri*, the presence of astragalus honey was able to increase the DPPH scavenging activity by almost six times respect to the control (83.7% vs. 15.04%, respectively); in the case of *L. paracasei* and *L. rhamnosus*, this honey practically doubled their antioxidant capacity (63.29% and 52.49%) compared to the relative controls. Sainfoin honey positively influenced the antioxidant capacity of almost all probiotic strains tested. In addition, in this case, *L. gasseri* seemed to receive the most significant benefits from sainfoin honey, so much so that its antioxidant activity (74.38%) was five times higher than that of the control (15.04%). This honey almost doubled the antioxidant capacity of *L. casei* Shirota (60.33% vs. 37.01% of the control) and exerted a beneficial effect on the antioxidant capacity of *L. plantarum* (41.78% vs. 29.09%). Carob, indigo, and alfalfa honeys exerted a weaker influence, limited to two of the five probiotic strains. Carob honey positively influenced the antioxidant activity of *L. gasseri* (35.04% vs. 15.04% of the control). Alfalfa honey increased the antioxidant activity of *L. gasseri* (41.4% vs. 15.04%) and *L. plantarum* (39.41% vs. 29.09%). Alfalfa honey increased the antioxidant activity of *L. gasseri* nearly 3-fold (41.4%). Indigo honey, unlike the other honeys, did not seem able to influence the antioxidant capacity of *L. gasseri*, and had a positive effect on the antioxidant capacity of *L. paracasei* (52.89% vs. 31.03% of the control), and that of *L. casei* Shirota (46.57% vs. 37.31% of the control). The OH-scavenging activity of the honeys was much more evident in the test conducted by measuring the ability of the probiotic bacterial cells to act on the hydroxyl radicals produced by the Fenton reaction occurring between H_2_O_2_ and FeSO_4_. As shown in Table 1b) only in few cases (for example, for sainfoin, carob, and alfalfa honey on *L. plantarum*, and for indigo honey on *L. plantarum* and *L. gasseri*), the honey, added to the culture medium as a substitute of glucose, did not improve the effects of the control. Once again, astragalus honey proved to be the most effective, improving the OH-scavenging capacity of all the probiotic strains, with percentages consistently higher than those exhibited by the control and which, in the test conducted on *L. paracasei* cells, succeeded in increasing its effectiveness almost six-fold. Compared to the OH-scavenging capacity values of the control, *L. rhamnosus* was the probiotic strain that received the most significant benefits from the presence of the honey, so much so that, in the presence of astragalus honey, its OH-scavenging capacity increased from 5.07% to 33.15%. In any case, even in the presence of other honeys, such a capacity was never less than 17.45% (when it was grown in the presence of carob honey). LAB exhibited a different behavior in the ILAP test (Table 1c). This test is utilized to measure lipid oxidation and antioxidant activity in physiological systems; linoleic acid (an essential fatty acid) is used as substrate; the possible linoleic acid peroxide has pathological consequences. In fact, lipid peroxidation in living cells is associated with serious damage to essential structural proteins and enzymes. The data of the ILAP test show that the honeys did not influence *L. paracasei*: when its growth was carried out in the presence of sainfoin, indigo, and carob honey; the inhibition of lipid peroxidation was close to or even equal to zero indeed. As regards *L. casei* Shirota, although the presence of honey determined a certain efficacy in inhibiting of lipid peroxidation, this was consistently lower than that exhibited by the probiotic strain grown in MRS with glucose. *L. rhamnosus* and *L. plantarum* showed better results than the respective controls only when they grew in the presence of carob (*L. rhamnosus*) and sainfoin honey (*L. plantarum*). Once again, *L. gasseri* proved to be the probiotic that benefited the most from the presence of honey, in particular carob (31.10%), alfalfa (28.66%) and sainfoin honey (20.44%). In the RP test (Table 1d), *L. paracasei* and *L. gasseri* were influenced by the honeys in a somewhat non-specific manner; indigo honey did not influence the power of probiotics, in particular of *L. casei* Shirota, *L. rhamnosus* and above all, *L. plantarum*. Conversely, astragalus honey strongly ameliorated the activity of the bacterial cells, with values not lower than 0.709 mM cysteine (in the case of *L. rhamnosus*), to 1.177 mM cysteine (*L. casei* Shirota). 

By comparing the antioxidant activity of the lactic acid bacteria grown in the presence of the studied honeys with the relative controls (rate sample/control, s/c), for each test carried out, we tried to identify whether and which type of honey could have influenced the bacterial antioxidant performances. All the honeys were able to increase the hydroxyl scavenging efficacy, so much so that the ratio between the test results conducted in the presence of honey and the control (in the presence of glucose) was never less than 3.44. Astragalus honey undoubtedly increased the hydroxyl scavenging efficacy of *L. paracasei* (sample/control = 5.52) and *L. rhamnosus* (s/c = 6.53). Astragalus and sainfoin honeys significantly increased the DPPH scavenging activity of *L. gasseri* (s/c = 5.56 and 4.94, respectively). The analysis of the s/c ratios show that, with respect to the hydroxyl scavenging activity, in the test conducted with DPPH, *L. plantarum* seemed capable of exerting more DPPH scavenging action than hydroxyl scavenging activity, so much so that, in the presence of sainfoin, efficacy increased 3-fold (s/c = 1.43 vs. 0.39, respectively) and almost seven-fold when *L. plantarum* grew in the presence of carob honey (s/c = 1.35 and 0.28, respectively). Instead, astragalus honey causes a slight lowering of DPPH scavenging activity than the hydroxyl scavenging activity (s/c= 1.12 and 1.41, respectively). Astragalus honey also greatly influenced the reducing power of probiotic bacteria cells, with ratio values that—compared to the control—were also found to be 5.04 (*L. plantarum*), 3.63 (*L. gasseri*), 3.18 (*L. paracasei*), 2.37 (*L. rhamnosus*), and 1.36 (*L. casei* Shirota). Compared to the other tests, astragalus honey positively influenced the RP of *L. plantarum* (s/c = 5.04); alfalfa honey (s/c = 2.08) also exhibited a good influence. In the ILAP test, contrary to what we observed for the other tests, carob (s/c = 8.17), alfalfa (s/c= 7.59), and sainfoin honey (s/c = 5.36) significantly influenced the inhibitory capacity of *L. gasseri* cells on lipid peroxidation. Astragalus honey, which was also able to influence the antioxidant efficacy of *L. gasseri* in the other tests, was less effective in the ILAP test but still able to double the effectiveness of this bacterium compared to the control (s/c = 1.98). Among the other bacterial strains, only *L. rhamnosus* (s/c = 4.83) seemed, in the ILAP test, to be positively influenced by the presence of carob honey. *L. plantarum* seemed to benefit above all from the presence of sainfoin honey in the culture medium (s/c = 1.21). Thus, the antioxidant activity, which is generally exhibited by many probiotics, including lactobacilli, to inhibit or limit the production of oxidant compounds in the intestine and thus hypothetically interfering with pathologies such as colon cancer [50] can, in many cases, be enhanced by the presence, in their growth environment, of sugar sources, such as, in our experiments, the tested honeys. Our future work will be to analyze the bioactive compounds produced by probiotic strains during honey fermentation, able to reduce oxidative stress, limiting or preventing the formation of ROS [51], to exert antioxidant activity by chelating metal ions and to exhibit reducing capacity [52]. The fact that, in several cases, the LABs have improved their ability to inhibit linoleic acid peroxidation is undoubtedly an exciting sign of the increased functional capacity of probiotics, albeit in vitro. Previous studies demonstrated that such ability makes the probiotics able to alleviate oxidative and metabolic damage in the liver, as evidenced for example for *L. plantarum* As1 [53], also in animal models [54].

### 3.3. Anti-Bacterial Activity of the Legumes’ Honey

Monofloral bioactive honeys are highly sought after and priced accordingly [55], as seen in the growing global demand for specialist pharmaceutical honeys such as the manuka (*Leptospermum scoparium* J. R. Forst & G. Forst) honey, which is one of the most famous worldwide. There is also a strong impulse to exploit various honeys, including leguminous ones, as medicinal honeys [56,57]. For thousands of years, honey, including some types of monofloral honey obtained from legume flowers, has also received considerable attention for its therapeutic properties, mainly due to its capacity to act against bacteria, yeasts, and fungi [58,59,60,61]. Table 2 shows the growth of the pathogenic strains in the presence of 40 and 80 μL/mL of the supernatants of the probiotics incubated in MRS. 

The choice to evaluate the impact of the honey on such pathogens is due to their role in human health. *A. baumannii* is commonly found in soil and water. It is also known to colonize human skin and can cause various infections, particularly in people with weakened immune systems or hospital settings. Moreover, this bacterium has become increasingly resistant to many commonly used antibiotics, making it difficult to treat infections, mainly due to its ability to acquire resistance genes and adapt to different environments [62]. The danger and diffusion of *A. baumanni* at the nosocomial level has grown a lot in recent times, concurrently with its increased antibiotic resistance [63]. The toxinogenic *E. coli*, has the capacity to produce toxins, and can cause severe foodborne illness [64]. *L. monocytogenes* can cause a severe foodborne illness called listeriosis. It can contaminate a variety of foods, such as raw and undercooked meat and poultry, unpasteurized dairy products, and ready-to-eat foods, including deli meats, hot dogs, and soft cheeses [65]. *P. aeruginosa* can cause many infections, particularly in people with weakened immune systems. It is known for its antibiotic resistance, making the infections caused by this bacterium difficult to treat [66]. *S. aureus* can cause a range of infections, from minor skin infections to more severe infections such as pneumonia, sepsis, and endocarditis. Methicillin-resistant *S. aureus* (MRSA) is resistant to many antibiotics, making it difficult to treat. MRSA infections can be hazardous for individuals with weakened immune systems, such as those in hospitals or nursing homes [67]. 

The replacement of glucose as an energy source with the five honeys during the growth of probiotic bacteria determined a change in bacterial metabolism. The growth supernatants of the probiotic bacteria, exhibited a different antibacterial efficacy, compared to the control (represented by the supernatants of the probiotic bacteria grown in MRS, Table 3), according to the type of honey, to the pathogen on which tests of antibacterial activity, as well as to the probiotic bacterium grown under those conditions.

In general, astragalus and carob honey were very effective, with an antibacterial performance that, compared to the control, increased up to 41.19% (supernatant of *L. plantarum* grown in the presence of astragalus honey vs. *E. coli*) and 53.94% (supernatant of *L. paracasei* grown in the presence of carob honey tested vs. *A. baumannii*). The activity of the honeys also changed regarding the probiotic strain. Thus, for example, astragalus honey did not increase the antibacterial vigor of *L. rhamnosus* (except for an apparent increase in antibacterial activity, +24.20% vs. *S. aureus*). Instead, all the other honeys, used as an alternative energy source to glucose, reinforced the antibacterial strength of the *L. rhamnosus* supernatant after growth vs. all the pathogenic strains, with percentages higher than those exerted by the respective controls also by 38.34% and 36.23% (vs. *E. coli*, when we used supernatants from alfalfa and carob honey growth media, respectively). On the contrary, the presence of sainfoin honey in the growth medium of *L. rhamnosus*, although It increased the antibacterial efficacy of the relative supernatant vs. *S. aureus* (+26.55% compared to the control). However, it was not able to reinforce its antibacterial action vs. the other pathogenic strains.

In contrast, carob, alfalfa, and indigo honeys increased the antibacterial efficacy against almost all pathogens. The tested honeys did not increase the antibacterial effectiveness of the supernatants compared to the control, except for a slight increase observed when we tested the supernatant of *L. gasseri* grown with alfalfa honey. Table 4 shows the MIC of the honey, determined to evaluate subsequently their inhibitory action on the bacterial ability to form a biofilm.

The results showed that, with the sole exception of the indigo honey (ineffective vs. *A. baumannii* and *P. aeruginosa*), the honeys inhibited all the tested bacteria used as models of pathogens (Table 5).

The inhibition of biofilm formation ranged between 1.02 (10 µg/mL of indigo honey vs. *S. aureus*) and 81.71% (20 µg/mL of carob honey vs. *E. coli*). *E. coli* was susceptible to all honeys, with a biofilm inhibitory activity ranging between 35.70 and 81.71% (with 20 µg/mL) and never less than 26.62% (10 µg/mL of indigo honey). The honeys inhibited the biofilm formation of *L. monocytogenes*. Except for indigo honey (whose activity was irrelevant at 10 µg/mL), this strain was generally sensitive to the honeys, with an inhibitory activity reaching 57.88% (with 20 µg/mL of astragalus honey) and resulting never less than 40.92% (with 20 µg/mL of indigo honey). *P. aeruginosa* resulted insensitive to almost all kinds of honey, with the exception, as previously mentioned, of indigo honey and, to a less extent, to the action of sainfoin honey, which inhibited the formation of its biofilm, albeit in a weak way (inhibition = 15.36%). S. aureus was sensitive to the action of all honeys, with inhibition not less than 28.06% (20 µg/mL of alfalfa honey), up to 56.22% (20 µg/mL of carob honey). Alfalfa honey managed to inhibit the biofilms of all five pathogens, with percentages of inhibition reaching 49.91% (20 µg/mL vs. *E. coli*). The anti-biofilm activity exhibited by the five honeys confirmed the effectiveness that honey, in general, can have in blocking or at least limiting those structural and metabolic changes that induce pathogenic bacteria to increase their virulence. Abbas [68] showed an apparent efficacy of Manuka honey and, to a lesser extent, of clover honey, in exerting an anti-biofilm action against five clinical isolates of *S. aureus*, *P. aeruginosa*, *Klebsiella* spp., and *Proteus mirabilis* isolated from diabetic foot ulcers, suggesting that they could help combat these pathologies. Among the leguminous kinds of honey, clover honey, also due to the abundance of vegetable material that bees can use, is undoubtedly the most studied and, in many respects, can be considered an exciting support for traditional medicine [56].

Honey, particularly from black locust, linden, and sunflower, could fight the most common biofilm-forming respiratory tract pathogens, such as *Haemophilus* spp., *P. aeruginosa*, and *Streptococcus pneumoniae* [69]. Given the considerable worldwide diffusion of the Fabaceae and the significant applications of their honey, various studies demonstrated that the pollen and nectar of these plants are sources of compounds with high nutraceutical value [70]. Moreover, the honey recovered from these plants could have, mainly in the case of clover honey, also the capability to affect the growth and virulence of pathogenic bacteria [71]. The evaluation of the activity of honey against biofilms is significant from a pre-clinical viewpoint demonstrating that honey can both inhibit biofilm formation and reduce the viability of established, mature biofilms, as shown in previous studies [17,72].

### 3.4. Antibiofilm Activity of Probiotic Growth-Medium Containing Legumes’ Honey Instead of Glucose

Several studies ascertained a prebiotic effect of honey when administered to different types of populations. This effect also translates into the influence that honey can have on the pathogenic or unwanted microorganisms constituting the bacterial flora of the organism. Aly et al. [73] have shown that supplementing children with a milk formula containing honey can lower the concentration of unwanted bacteria such as Enterobacter. At the same time, it can increase the population of bifidobacteria and lactobacilli. At least to our knowledge, there is no work reporting the effect of honey, and mainly of legumes honey, on the anti-biofilm capacity of the probiotic growth medium, in which honey substituted the conventional glucose as energy font. Thus, we also evaluated this aspect, determining the anti-biofilm efficacy of the growth supernatants of the probiotic strains incubated in MRS in which honey replaced glucose. Results are shown in Table 6.

Based on the results of the antibacterial activity, we used the same amount of supernatant also to assay the anti-biofilm activity calculating, as percentage, the values with respect to the biofilm formation of untreated pathogenic bacteria. As shown in Table 6, the presence of glucose did not give to the growth supernatants of probiotic bacteria the capacity of acting on the formation of pathogenic biofilms, with few exceptions. Only the supernatant of *L. gasseri* (with inhibition until 47.97% vs. *S. aureus*) and the supernatant of *L. rhamnosus* (with 79.38% inhibition vs. *A. baumannii*) showed an apparent antibiofilm efficacy. The presence of honey in the LAB growth medium influenced its antibiofilm action. Astragalus honey significantly influenced the antibiofilm activity of the LAB supernatants, especially in the tests conducted vs. *L. monocytogenes* and *S. aureus*. In this case, the inhibition by the supernatants reached percentages of over 45% (supernatant of *L. casei* Shirota vs. *L. monocytogenes*; supernatant of *L. rhamnosus* vs. *L. monocytogenes*; supernatants of *L. gasseri*, *L. plantarum* and *L. rhamnosus* vs. *S. aureus*). The different behavior exhibited by the supernatants can be seen above all by considering the antibiofilm activity shown by honey as it is against the same pathogenic strains. In the case of astragalus, its fermentation by the various probiotic strains gave rise to a different behavior. Thus, as already reported, the supernatant of *L. casei* Shirota was active against *L. monocytogenes*; the supernatants of *L. gasseri* and *L. plantarum* acted both against the biofilm of *L. monocytogenes* and *S. aureus*, the supernatant of *L. plantarum* exhibited an evident antibiofilm activity only vs. *S. aureus*. The presence of carob honey as an energy source instead of glucose determined the efficacy of antibiofilm action also against *E. coli*, with inhibition percentages that, in some cases, were higher than 30% (the supernatant of *L. casei* Shirota, *L. gasseri*, and *L. plantarum* with inhibition of 32.13, 36.97 and 38.58%, respectively).

Furthermore, all the supernatants could inhibit the biofilm of *L. monocytogenes*, with percentages in some cases higher than 40% (*L. gasseri* and *L. plantarum*). In contrast, the supernatant of *L. paracasei* grown in the presence of that honey showed an inhibitory effect only of 12.16%. Unlike the results of tests conducted with the supernatants in which astragalus replaced glucose, the fermentation process in the presence of carob honey made the supernatant of *L. casei* more effective vs. *S. aureus* compared to the supernatant of *L. casei* Shirota grown in the presence of astragalus (27.32 and 7.06%, respectively). In contrast, the *L. rhamnosus* supernatant grown in the presence of astragalus honey exhibited antibiofilm efficacy vs. *S. aureus* undoubtedly more significant than that shown by the supernatant of *L. rhamnosus* after growth in the presence of carob honey, which in this case proved to be ineffective.

The supernatant of *L. gasseri* was the only one to show a specific inhibitory strength (33.05%) against *A. baumannii* (which instead had been inhibited with an inhibitory potency of over 70% in the test conducted with the supernatant of *L. rhamnosus* grown in MRS with glucose). Only when the supernatants gave rise from the growth in the presence of sainfoin honey we noted a certain inhibition vs. *A. baumannii*, particularly with the supernatant of *L. plantarum* and *L. rhamnosus* (30.42 and 36.25%, respectively). The supernatants of the probiotic strains grown in the presence of alfalfa honey could limit the biofilm formed by *L. monocytogenes* in a not-so-effective way. The supernatant of *L. plantarum* (inhibiting the biofilm of this pathogen by 60.05%), and particularly the supernatant of *L. rhamnosus* (whose inhibitory efficacy was able to limit the formation of the biofilm of *L. monocytogenes* up to 76.59%) represented the only two exceptions. The supernatants of the probiotic strains grown in the presence of alfalfa honey were effective in inhibiting the biofilm of S. aureus, in particular, the supernatant of *L. gasseri* (46.04% inhibition) and *L. paracasei* (31.81% inhibition). The supernatants of L. plantarum (19.19% inhibition) and *L. casei* Shirota (14.75% inhibition) were effective. The supernatant of *L. rhamnosus* seemed more effective. The supernatants of the probiotics grown in the presence of indigo honey were usually ineffective in inhibiting the biofilm, with few important exceptions.

Conversely, the supernatant of *L. casei* Shirota was usually effective and, in the test carried out against *P. aeruginosa*, exhibited an anti-biofilm vigor reaching 79.67%. Therefore, this represented the only case of anti-biofilm activity shown vs. *P. aeruginosa* in all the tests performed with the probiotic growth supernatants. Thus, the fermentation of the five honeys by probiotics led them to lose their antibiofilm efficacy against this pathogen, which was evident in the test conducted with all honeys before fermentation. In that case, legumes’ honey (except indigo honey) contrasted the *P. aeruginosa* biofilm, with inhibition ranging between 15.36 (sainfoin honey) and 51.20% (carob honey). The growth supernatants of the various probiotics acted against the formation of the biofilm of *S. aureus*, with efficacy ranging from 13.10 (supernatant of *L. casei* Shirota) to 79.99% (supernatant of *L. paracasei*). They were often ineffective vs. *E. coli* (except supernatant of *L. casei* Shirota). A moderate inhibitory effect was also exerted vs. *L. monocytogenes*, with inhibition that reached 36.53% (in the case of the test conducted with the supernatant of *L. casei* Shirota). Some supernatants of probiotics grown in the presence of sainfoin honey—in particular that of *L. plantarum* and that of L. rhamnosus- exhibited inhibitory efficacy vs. *A. baumannii* (inhibition of 30.42 and 36.25%, respectively). On the other hand, they were ineffective vs. *E. coli* and *L. monocytogenes* (except for a slight biofilm inhibitory action exhibited by the supernatant of *L. casei* vs. *L. monocytogenes*).

## 4. Conclusions

The emergence of resistant bacteria leads researchers to find new strategies to fight them. The honeys described in our study showed significant inhibitory effects against five of the most dangerous pathogenic bacteria, acting on their mature biofilm, a more complex situation to fight, particularly worrying for particular population segments, such as infants and older people. Future work will be addressed to the biochemical characterization of the legumes’ honey and to exploit how its biochemistry can influence other biological properties, both of the product itself and of the probiotics grown in their presence. Studies are in progress to characterize some classes of molecules, such as polyphenols and volatile compounds, present in these five types of honey and giving also rise from pollen and nectar [73], which presence is related to the geographical area of origin and to the botanical species from which it derives [74], and that can affect the biological properties of the honey, including the antibacterial and prebiotic activity [75]. These results represent, also the basis for further study about the discovery and study of single molecules or their mixtures with post-biotic activity, deriving from the fermentation of polyphenols and volatile compounds present in honey used by probiotics as an energy source, and that could provide health benefits to the host [76,77,78]. Therefore, such aspect has been recently studied in other bee products and bacteria isolated by the bee [79,80] and can be applicable to conventional probiotics, such as *Lacticaseibacillus rhamnosus* or *Lacticaseibacillus paracasei* [81], two of the five strains of probiotics studied in the present work. Finally, identifying the molecules able to most influence the antibiofilm efficacy, in honey and after its fermentation, will also help to study what could be the mechanisms that allow these molecules to limit the pathogens virulence. A similar consideration will concern the identification of the molecules produced by honey fermentation capable of influencing to a greater or lesser extent (and in different way) the antioxidant activity of probiotic bacteria, and therefore also capable of exhibiting post-biotic properties.

## Figures and Tables

**Figure 1 foods-12-03338-f001:**
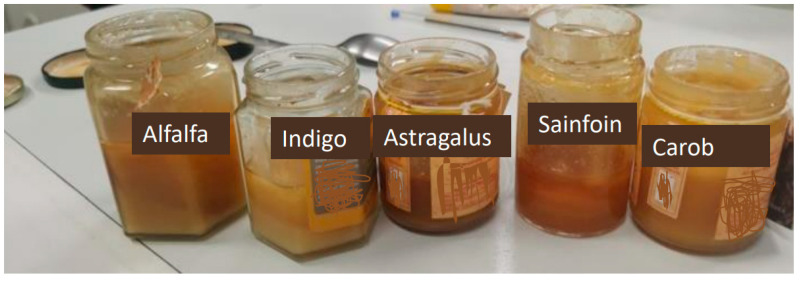
The commercial organic legumes’ honey used for the experiments.

**Figure 2 foods-12-03338-f002:**
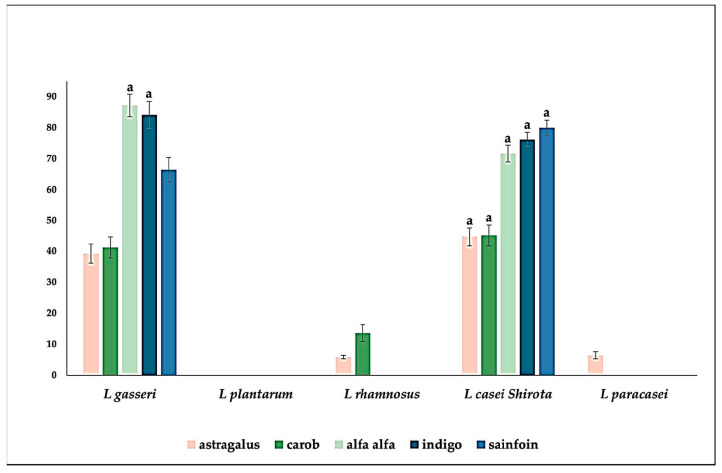
Differences of the growth of lactic acid bacteria in presence of the legumes’ honey, evaluated as percentage (on *Y*-axis) respect to control. Data are the average of three independent experiments ± SD. ^a^ *p* < 0.5 (ANOVA followed by Dunnett’s multiple comparison test).

**Figure 3 foods-12-03338-f003:**
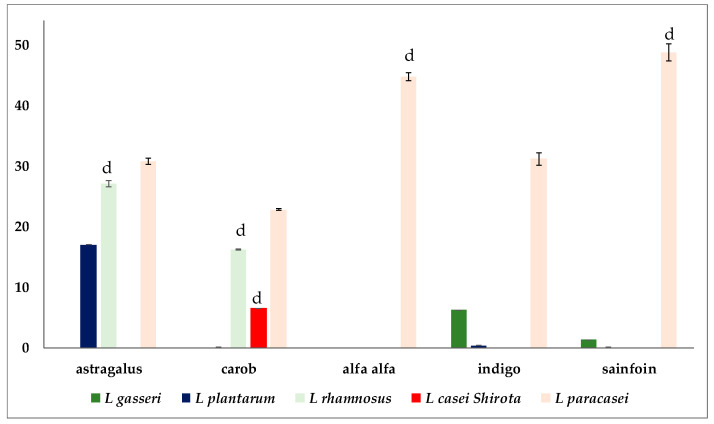
The in vitro differences of the adhesive capacity of lactic acid bacteria in presence of the honeys, evaluated as percentage (on *Y*-axis) respect to the control. Data are the average of three independent experiments ± SD. ^d^ *p* < 0 (ANOVA followed by Dunnett’s multiple comparison test).

**Table 1 foods-12-03338-t001:** (panels a–d) Antioxidant capacity exhibited by the probiotic strains *L. paracasei*, *L. gasseri*, *L. casei Shirota*, *L. rhamnosus*, *L. plantarum,* in the presence of different types of legumes’ honey.

**(a) DPPH**						
	Alfalfa	Astragalus	Carob	Indigo	Sainfoin	MRS
*L. paracasei*	32.22 ± 0.72 ^ns^	63.29 ± 0.39 ^b^	15.25 ± 1.54 ^b^	52.89 ± 0.86 ^b^	16.49 ± 2.68 ^b^	31.03 ± 1.27
*L. gasseri*	41.4 ± 0.81 ^c^	83.7 ± 1.37 ^d^	35.04 ± 9.02 ^c^	13.11 ± 1.55 ^ns^	74.38 ± 0.29 ^d^	15.04 ± 1.24
*L. casei Shirota*	34.12 ± 0.83 ^ns^	32.74 ± 2.04 ^ns^	15.13 ± 2.41 ^b^	46.57 ± 1.36 ^a^	60.33 ± 0.55 ^b^	37.31 ± 2.44
*L. rhamnosus*	23.07 ± 0.89 ^ns^	52.49 ± 1.03 ^b^	20.08 ± 2.82 ^ns^	13.24 ± 3.84 ^a^	18.46 ± 1.41 ^a^	24.33 ± 1.18
*L. plantarum*	17.1 ± 0.7 ^b^	32.71 ± 2.26 ^ns^	39.41 ± 2.92 ^b^	11.91 ± 2.69 ^b^	41.78 ± 2.44 ^b^	29.0 ± 3.89
**(b) OH radical scavenging activity**						
	Alfalfa	Astragalus	Carob	Indigo	Sainfoin	MRS
*L. paracasei*	13.31 ± 3.31 ^b^	41.11 ± 4.99 ^d^	10.39 ± 6.52 ^b^	22.53 ± 0.02 ^c^	18.29 ± 1.95 ^c^	7.44 ± 1.54
*L. gasseri*	28.71 ± 5.1 ^c^	36.60 ± 5.67 ^d^	22.83 ± 5.71 ^b^	6.09 ± 6.07 ^c^	36.25 ± 6.64 ^d^	14.40 ± 0.73
*L. casei Shirota*	19.92 ± 4.05 ^ns^	32.21 ± 9.25 ^c^	19.04 ± 0.87^ns^	22.66 ± 4.4^nd^	26.01 ± 6.89 ^b^	18.68 ± 5.75
*L. rhamnosus*	20.63 ± 7.46 ^c^	33.16 ± 7.45 ^d^	17.46 ± 2.11 ^c^	22.46 ± 5.09 ^c^	20.85 ± 1.46 ^c^	5.07 ± 3.4
*L. plantarum*	8.93 ± 2.61 ^c^	39.5 ± 6.19 ^b^	7.88 ± 1.08 ^c^	7.00 ± 3.62 ^c^	11.07 ± 4.59 ^c^	27.98 ± 7.05
**(c) Inhibitory activity of LAB cells lipidic peroxidation with liposomial system**						
	Alfalfa	Astragalus	Carob	Indigo	Sainfoin	MRS
*L. paracasei*	2.54 ± 0.48 ^a^	1.25 ± 0.42 ^a^	0 ± 0	0.62 ± 0.11 ^a^	0.54 ± 0.14 ^a^	5.06 ± 0.23
*L. gasseri*	28.66 ± 1.21 ^d^	7.57 ± 1.31 ^a^	31.11 ± 2.55 ^d^	1.45 ± 0.32 ^a^	20.44 ± 3.45 ^c^	3.81 ± 0.75
*L. casei Shirota*	8.63 ± 1.65 ^c^	4.3 ± 0.26 ^c^	5.47 ± 0.14 ^c^	8.21 ± 1.05 ^c^	13.75 ± 1.75 ^b^	27.41 ± 0.05
*L. rhamnosus*	1.72 ± 0.15 ^ns^	0 ± 0	11.89 ± 1.74 ^b^	1.95 ± 0.05 ^ns^	1.83 ± 0.33 ^ns^	2.46 ± 0.97
*L. plantarum*	1.17 ± 0.72 ^ns^	1.46 ± 0.03 ^ns^	0.5 ± 0.05 ^ns^	0 ± 0	2.03 ± 0.03 ^ns^	1.67 ± 0.29
**(d) CFS reduction activity**						
	Alfalfa	Astragalus	Carob	Indigo	Sainfoin	MRS
*L. paracasei*	0.389 ± 0.05 ^a^	0.968 ± 0.04 ^c^	0.531 ± 0.04 ^a^	0.43 ± 0.01 ^a^	0.549 ± 0.04 ^a^	0.304 ± 0.01
*L. gasseri*	0.593 ± 0.03 ^b^	0.949 ± 0.02 ^c^	0.669 ± 0.03 ^b^	0.104 ± 0.02 ^a^	0.732 ± 0.03 ^b^	0.261 ± 0.01
*L. casei Shirota*	0.604 ± 0.04 ^a^	1.177 ± 0.04 ^b^	0.203 ± 0.02 ^b^	0.461 ± 0.01 ^b^	0.835 ± 0.03 ^ns^	0.863 ± 0.03
*L. rhamnosus*	0.35 ± 0.02 ^ns^	0.709 ± 0.02 ^b^	0.328 ± 0.02 ^ns^	0.216 ± 0.05 ^ns^	0.191 ± 0.02 ^ns^	0.298 ± 0.02
*L. plantarum*	0.357 ± 0.01 ^a^	0.863 ± 0.03 ^c^	0.092 ± 0.04 ^a^	0.026 ± 0.02 ^c^	0.131 ± 0.01 ^ns^	0.171 ± 0.01

Data are the average of three independent experiments ± SD. ^a^: *p* < 0.5; ^b^ *p* < 0.01; ^c^ *p* < 0.001; ^d^ *p* < 0.0001 compared with the control in MRS. ns: not significative, compared with the average values (ANOVA followed by Dunnett’s multiple comparison test).

**Table 2 foods-12-03338-t002:** Antibacterial activity of the supernatants of the five LAB grown in MRS performed with the supernatant (40 μL/mL and 80 μL/mL) of the growth medium previously filtered. The data are expressed as OD (read at λ = 600 nm).

	*Acinetobacter baumanii*	*Escherichia* *coli*	*Listeria monocytogenes*	*Pseudomonas aeruginosa*	*Staphylococcus aureus*
*L. casei* 40 μL/mL	0.3683 ^b^ ± 0.017	0.3164 ^a^ ± 0.043	0.6587 ^c^ ± 0.044	0.3964 ^c^ ± 0.019	0.5063 ^b^ ± 0.049
*L. casei* Shirota 80 μL/mL	0.3538 ^b^ ± 0.018	0.3152 ^a^ ± 0.021	0.5315 ^c^ ± 0.049	0.369 ^c^ ± 0.044	0.368 ^c^ ± 0.067
*L. gasseri* 40 μL/mL	0.4178 ^a^ ± 0.056	0.3216 ^a^ ± 0.091	0.832 ^b^ ± 0.077	0.4214 ^b^ ± 0.021	0.4807 ^b^ ± 0.026
*L. gasseri* 80 μL/mL	0.3198 ^b^ ± 0.023	0.2563 ^b^ ± 0.057	0.7849 ^b^ ± 0.28	0.352 ^c^ ± 0.20	0.4021 ^b^ ± 0.024
*L. paracasei* subsp. *paracasei* 40 μL/mL	0.4082 ^a^ ± 0.028	0.2579 ^b^ ± 0.033	1.0114 ± 0.064	0.4218 ^b^ ± 0.051	0.4834 ^b^ ± 0.042
*L. paracasei* subsp. *paracasei* 80 μL/mL	0.3863 ^b^ ± 0.019	0.249 ^b^ ± 0.031	0.5926 ^c^ ± 0.057	0.3457 ^c^ ± 0.017	0.4216 ^b^ ± 0.023
*L. plantarum* 40 μL/mL	0.3436 ^b^ ± 0.027	0.4001 ± 0.032	0.8792 ^b^ ± 0.076	0.3625 ^c^ ± 0.019	0.4903 ^b^ ± 0.026
*L. plantarum* 80 μL/mL	0.3403 ^b^ ± 0.057	0.2541 ^b^ ± 0.054	0.7526 ^b^ ± 0.061	0.3327 ^c^ ± 0.038	0.4544 ^b^ 0.034
*L. rhamnosus* 40 μL/mL	0.5211 ± 0.044	0.4371 ± 0.019	1.1112 ± 0.093	0.4995 ^b^ ± 0.067	0.5894 ^a^ ± 0.052
*L. rhamnosus* 80 μL/mL	0.5197 ± 0.048	0.3247 ± 0.032	0.7851 ^b^ ± 0.063	0.4214 ^b^ ± 0.038	0.5475 ^b^ ± 0.047
Control	0.5258 ± 0.041	0.4611 ± 0.035	1.093 ± 0.079	0.7229 ± 0.051	0.7583 ± 0.051

The data are the average of three independent experiments ± SD. ^a^ *p* < 0.5; ^b^ *p* < 0.01; ^c^ *p* < 0.001; compared with the positive control. ^a^ *p* < 0.5; ^b^ *p* < 0.01; ^c^ *p*< 0.001; compared with the average values (ANOVA followed by Dunnett’s multiple comparison test).

**Table 3 foods-12-03338-t003:** Comparison (in terms of percentage) of the antibacterial activity of the supernatants of lactic acid bacteria in the presence of astragalus (ASTR), carob (CAR), alfalfa (ALF), indigo (IND), and sainfoin (SAIN) honey, with respect to their antibacterial activity when they were grown in MRS.

ALF	Lc 40 μL/mL	Lc 80 μL/mL	Lg 40 μL/mL	Lg 80 μL/mL	Lpc 40 μL/mL	Lpc 80 μL/mL	Lpl 40 μL/mL	Lpl 80 μL/mL	Lrh 40 μL/mL	Lrh 80 μL/mL
*AB*	0 ± 0.00	0 ± 0.00	0 ± 0.00	0 ± 0.00	0 ± 0.00	4.91 ± 1.44	0 ± 0.00	0 ± 0.00	0 ± 0.00	17.70 ± 1.13
*EC*	1.35 ± 0.05	17.92 ± 1.70	23.44 ± 1.91	17.59 ± 1.65	4.53 ± 0.56	21.20 ± 1.75	41.43 ± 2.64	16.29 ± 1.17	36.23 ± 2.57	15.18 ± 1.13
*LM*	0 ± 0.00	0 ± 0.00	0 ± 0.00	5.80 ± 0.45	0 ± 0.00	0 ± 0.00	0 ± 0.00	0 ± 0.00	0 ± 0.00	0 ± 0.00
*PA*	0 ± 0.00	0.43 ± 0.12	22.66 ± 1.98	9.94 ± 0.88	0 ± 0.00	0 ± 0.00	0 ± 0.00	0 ± 0.00	0 ± 0.00	4.79 ± 0.04
*SA*	0 ± 0.00	0 ± 0.00	0 ± 0.00	12.33 ± 1.03	2.54 ± 0.44	9.36 ± 0.91	0 ± 0.00	22.02 ± 1.90	5.09 ± 0.45	24.23 ± 2.03
**ASTR**	**Lc** **40 μL/mL**	**Lc** **80** **μL/mL**	**Lg** **40** **μL/mL**	**Lg** **80** **μL/mL**	**Lpc** **40** **μL/mL**	**Lpc** **80** **μL/mL**	**Lpl** **40 μL/mL**	**Lpl** **80** **μL/mL**	**Lrh 40 μL/mL**	**Lrh 80** **μL/mL**
*AB*	0 ± 0.00	55.31 ± 4.41	0 ± 0.00	0 ± 0.00	0 ± 0.00	0 ± 0.00	0 ± 0.00	0 ± 0.00	0 ± 0.00	0 ± 0.00
*EC*	0 ± 0.00	26.17 ± 1.57	29.37 ± 2.45	39.11 ± 2.67	0.27 ± 0.02	18.51 ± 1.67	11.81 ± 1.13	41.19 ± 2.57	0 ± 0.00	4.71 ± 0.32
*LM*	0 ± 0.00	0 ± 0.00	0 ± 0.00	0 ± 0.00	0 ± 0.00	0 ± 0.00	0 ± 0.00	0 ± 0.00	0 ± 0.00	0 ± 0.00
*PA*	0 ± 0.00	19.02 ± 1.13	0 ± 0.00	11.90 ± 1.13	30.27 ± 2.85	31.47 ± 3.02	8.68 ± 0.57	34.41 ± 2.67	0 ± 0.00	0 ± 0.00
*SA*	0 ± 0.00	0 ± 0.00	0 ± 0.00	10.79 ± 1.13	0 ± 0.00	38.49 ± 3.45	0 ± 0.00	2.04 ± 0.04	0 ± 0.00	24.20 ± 1.84
**CAR**	**Lc** **40 μL/mL**	**Lc** **80** **μL/mL**	**Lg** **40** **μL/mL**	**Lg** **80** **μL/mL**	**Lpc** **40** **μL/mL**	**Lpc** **80** **μL/mL**	**Lpl** **40 μL/mL**	**Lpl** **80** **μL/mL**	**Lrh** **40 μL/mL**	**Lrh 80** **μL/mL**
*AB*	0 ± 0.00	0 ± 0.00	0 ± 0.00	29.17 ± 2.67	9.08 ± 0.45	53.94 ± 2.67	0 ± 0.00	10.10 ± 0.85	0 ± 0.00	22.16 ± 1.44
*EC*	15.96 ± 1.57	19.92 ± 1.84	30.84 ± 3.02	14.82 ± 1.13	3.87 ± 0.04	20.04 ± 1.67	40.11 ± 3.53	14.24 ± 1.15	38.34 ± 3.03	29.59 ± 2.01
*LM*	0 ± 0.00	0 ± 0.00	0 ± 0.00	0 ± 0.00	0 ± 0.00	0 ± 0.00	0 ± 0.00	0 ± 0.00	0 ± 0.00	0 ± 0.00
*PA*	9.68 ± 1.01	12.38 ± 1.13	0 ± 0.00	10.59 ± 0.84	0 ± 0.00	4.68 ± 3.44	0 ± 0.00	5.62 ± 0.47	0 ± 0.00	3.53 ± 0.13
*SA*	0 ± 0.00	0 ± 0.00	0 ± 0.00	4.37 ± 0.33	0 ± 0.00	10.05 ± 0.55	0 ± 0.00	13.77 ± 1.13	8.29 ± 0.57	9.95 ± 1.05
**IND**	**Lc** **40 μL/mL**	**Lc** **80** **μL/mL**	**Lg** **40** **μL/mL**	**Lg** **80** **μL/mL**	**Lpc** **40** **μL/mL**	**Lpc** **80** **μL/mL**	**Lpl** **40 μL/mL**	**Lpl** **80** **μL/mL**	**Lrh 40 μL/mL**	**Lrh 80** **μL/mL**
*AB*	0 ± 0.00	0 ± 0.00	0 ± 0.00	0 ± 0.00	0 ± 0.00	0 ± 0.00	0 ± 0.00	0 ± 0.00	3.53 ± 0.13	14.60 ± 1.57
*EC*	1.99 ± 1.57	26.55 ± 1.67	8.73 ± 0.57	0 ± 0.00	0 ± 0.00	5.06 ± 044	17.54 ± 1.53	0 ± 0.00	26.12 ± 1.57	4.18 ± 0.34
*LM*	0 ± 0.00	0 ± 0.00	0 ± 0.00	0 ± 0.00	0 ± 0.00	0 ± 0.00	0 ± 0.00	0 ± 0.00	0 ± 0.00	0 ± 0.00
*PA*	0 ± 0.00	0 ± 0.00	3.13 ± 0.1)	0 ± 0.00	13.44 ± 0.12	0.23 ± 0.02	0 ± 0.00	0 ± 0.00	0 ± 0.00	0.42 ± 0.03
*SA*	6.67 ± 0.32	0 ± 0.00	1.62 ± 0.34	0 ± 0.00	1.73 ± 0.24	4.05 ± 0.34	13.78 ± 1.13	7.92 ± 0.45	0 ± 0.00	23.39 ± 1.67
**SAIN**	**Lc** **40 μL/mL**	**Lc** **80** **μL/mL**	**Lg** **40** **μL/mL**	**Lg** **80** **μL/mL**	**Lpc** **40** **μL/mL**	**Lpc** **80** **μL/mL**	**Lpl** **40 μL/mL**	**Lpl** **80** **μL/mL**	**Lrh 40 μL/mL**	**Lrh 80** **μL/mL**
*AB*	0 ± 0.00	2.71 ± 0.03	0 ± 0.00	0 ± 0.00	0 ± 0.00	0 ± 0.00	0 ± 0.00	0 ± 0.00	0 ± 0.00	0.19 ± 0.02
*EC*	10.14 ± 0.12	26.87 ± 1.67	16.30 ± 1.13	27.39 ± 2.04	0 ± 0.00	12.04 ± 1.0)	1.81 ± 1.07	35.94 ± 2.67	0 ± 0.00	0 ± 0.00
*LM*	0 ± 0.00	0 ± 0.00	0 ± 0.00	0 ± 0.00	0 ± 0.00	0 ± 0.00	0 ± 0.00	0 ± 0.00	0 ± 0.00	0 ± 0.00
*PA*	0 ± 0.00	0 ± 0.00	0 ± 0.00	5.01 ± 0.44	0 ± 0.00	0 ± 0.00	0 ± 0.00	0 ± 0.00	0 ± 0.00	0 ± 0.00
*SA*	0 ± 0.00	2.94 ± 0.09	6.98 ± 0.57	10.50 ± 0.45	0 ± 0.00	8.20 ± 0.78	5.83 ± 0.57	16.24 ± 1.13	0 ± 0.00	26.55 ± 2.05

Data are the average of three independent experiment ± SD. Lc: *Lacticaseibacillus casei* Shirota; Lg: *Lactobacillus gasseri*; Lpc: *Lacticaseibacillus paracasei* subsp. paracasei; Lpl: *Lactiplantibacillus plantarum*; Lrh: *Lacticaseibacillus rhamnosus*. *AB*: *A. baumannii*; *EC*: *E. coli*; *LM*: *L. monocytogenes*; *PA*: *P. aeruginosa*; *SA*: *S. aureus*.

**Table 4 foods-12-03338-t004:** Minimal inhibitory concentration (MIC, µg/mL) of the honeys, needed to block the metabolic activity of the five bacterial strains, evaluated through the resazurin test.

	*A. baumanii*	*E. coli*	*L. monocytogenes*	*P. aeruginosa*	*S. aureus*
Alfalfa	32 ± 1	34 ^d^ ± 2	35 ± 1	38 ± 4	35 ± 2
Astragalus	35 ^a^ ± 2	35 ^d^ ± 3	34 ± 2	40 ^b^ ± 2	40 ^b^ ± 4
Carob	40 ^d^ ± 3	30 ^a^ ± 2	34 ± 1	36 ± 2	35 ± 1
Indigo	50 ^d^ ± 2	34 ^d^ ± 2	40 ^d^ ± 2	40 ^b^ ± 3	45 ^d^ ± 3
Sainfoin	34 ± 2	32 ^b^ ± 2	32 ± 2	40 ^b^ ± 3	38 ± 2
Tetracycline	30 ± 2	25 ± 2	32 ± 1	34 ± 1	34 ± 1

Results are the average of three independent experiment ± SD. ^a^ *p* < 0.5; ^b^ *p* < 0.01; ^d^ *p* < 0.0001 compared with the positive control. (ANOVA followed by Dunnett’s multiple comparison test).

**Table 5 foods-12-03338-t005:** Inhibition of the biofilm formation by pathogenic bacteria by the astragalus (ASTR), carob (CAR), alfalfa (ALF), indigo (IND), and sainfoin (SAIN) honeys, calculated as percentage, assuming for the untreated bacteria an inhibition = 0.

	*Acinetobacter* *baumannii*	*Escherichia* *coli*	*Listeria monocytogenes*	*Pseudomonas* *aeruginosa*	*Staphylococcus* *aureus*
Alfalfa 10 μg/mL	5.64 ^a^ ± 0.15	45.61 ^d^ ± 2.52	0 ± 0.00	12.01 ^b^ ± 1.02	8.68 ^a^ ± 0.52
Alfalfa 20 μg/mL	40.58 ^d^ ± 3.67	49.91 ^d^ ± 3.45	23.42 ^c^ ± 1.78	37.61 ^d^ ± 3.52	28.85 ^c^ ± 1.45
Astragalus 10 μg/mL	26.45 ^d^ ± 1.32	32.38 ^d^ ± 1.22	49.84 ^d^ ± 2.13	20.25 ^c^ ± 1.05	26.01 ^c^ ± 2.02
Astragalus 20 μg/mL	37.47 ^d^ ± 2.09	35.70 ^d^ ± 2.45	57.88 ^d^ ± 3.04	29.27 ^d^ ± 2.02	39.52 ^d^ ± 2.78
Carob 10 μg/mL	0 ± 0.00	37.69 ^d^ ± 1.57	50.28 ^d^ ± 3.66	26.55 ^c^ ± 2.35	47.35 ^d^ ± 1.44
Carob 20 μg/mL	7.72 ^a^ ± 0.44	81.71 ^d^ ± 1.44	56.01 ^d^ ± 3.16	51.20 ^d^ ± 3.04	56.22 ^d^ ± 2.68
Indigo 10 μg/mL	0 ± 0.00	26.62 ^c^ ± 1.13	30.82 ^d^ ± 1.24	0 ± 0.00	1.02 ± 0.06
Indigo 20 μg/mL	0 ± 0.00	44.62 ^d^ ± 3.54	40.92 ^d^ ± 1.44	0 ± 0.00	28.06 ^c^ ± 2.04
Sainfoin 10 μg/mL	34.67 ^d^ ± 2.13	57.78 ^d^ ± 3.98	22.46 ^c^ ± 1.24	0 ± 0.00	40.03 ^d^ ± 2.16
Sainfoin 20 μg/mL	35.83 ^d^ ± 1.67	60.06 ^d^ ± 2.08	51.25 ^d^ ± 1.15	15.36 ^b^ ± 1.24	47.58 ^d^ ± 3.35

Results are the average of three independent experiment ± SD. ^a^ *p* < 0.5; ^b^ *p* < 0.01; ^c^ *p* < 0.001; ^d^ *p* < 0.0001 compared with the positive control. ^a^ *p* < 0.5; ^b^ *p* < 0.01; ^c^ *p* < 0.001; ^d^ *p* < 0.0001 compared with the average values (ANOVA followed by Dunnett’s multiple comparison test).

**Table 6 foods-12-03338-t006:** Inhibitory activity of 40 and 80 μμL/mL of supernatant of lactic acid bacteria grown in MRS or MRS without glucose but containing an equal amount of honey, against the biofilm formation of pathogenic strains.

MRS	Lc 40 μL/mL	Lc 80 μL/mL	Lg 40 μL/mL	Lg 80 μL/mL	Lpc 40 μL/mL	Lpc 80 μL/mL	Lpl 40 μL/mL	Lpl 80 μL/mL	Lrh 40 μL/mL	Lrh 80 μL/mL
*Acinetobacter* *baumannii*	0 ± 0.00	0 ± 0.00	0 ± 0.00	0 ± 0.00	0 ± 0.00	0 ± 0.00	0 ± 0.00	0 ± 0.00	0 ± 0.00	79.38 ^d^ ± 3.04
*Escherichia* *coli*	0 ± 0.00	0 ± 0.00	0 ± 0.00	0 ± 0.00	0 ± 0.00	0 ± 0.00	0 ± 0.00	0 ± 0.00	0 ± 0.00	0 ± 0.00
*Listeria* *monocytogenes*	0 ± 0.00	0 ± 0.00	2.89 ^a^ ± 0.57	3.30 ^a^ ± 0.13	0 ± 0.00	0 ± 0.00	0 ± 0.00	0 ± 0.00	0 ± 0.00	0 ± 0.00
*Pseudomonas* *aeruginosa*	0 ± 0.00	0 ± 0.00	0 ± 0.00	0 ± 0.00	0 ± 0.00	0 ± 0.00	0 ± 0.00	0 ± 0.00	0 ± 0.00	0 ± 0.00
*Staphylococcus* *aureus*	0 ± 0.00	0 ± 0.00	24.47 ^c^ ± 1.41)	47.97 ^d^ ± 3.57	0 ± 0.00	0 ± 0.00	0 ± 0.00	10.81 ^b^ ± 0.67	0 ± 0.00	6.28 ^a^ ± 0.57
**Alfalfa**	**Lc** **40 μL/mL**	**Lc** **80** **μL/mL**	**Lg** **40** **μL/mL**	**Lg** **80** **μL/mL**	**Lpc** **40** **μL/mL**	**Lpc** **80** **μL/mL**	**Lpl** **40 μL/mL**	**Lpl** **80** **μL/mL**	**Lrh 40 μL/mL**	**Lrh 80** **μL/mL**
*Acinetobacter* *baumannii*	0 ± 0.00	0 ± 0.00	0 ± 0.00	0 ± 0.00	0 ± 0.00	0 ± 0.00	0 ± 0.00	0 ± 0.00	0 ± 0.00	0 ± 0.00
*Escherichia* *coli*	0 ± 0.00	0 ± 0.00	0 ± 0.00	8.75 ^a^ ± 1.22	0 ± 0.00	0 ± 0.00	0 ± 0.00	18.89 ^b^ ± 0.67	0 ± 0.00	0 ± 0.00
*Listeria* *monocytogenes*	0 ± 0.00	12.91 ^b^ ± 1.24	0 ± 0.00	0 ± 0.00	0 ± 0.00	34.73 ^d^ ± 2.54	42.27 ^d^ ± 3.07	60.05 ^d^ ± 3.34	50.56 ^d^ ± 4.03	76.59 ^d^ ± 2.40
*Pseudomonas* *aeruginosa*	0 ± 0.00	1.44 ± 0.02	0 ± 0.00	0 ± 0.00	0 ± 0.00	0 ± 0.00	0 ± 0.00	0 ± 0.00	0 ± 0.00	0 ± 0.00
*Staphylococcus* *aureus*	0 ± 0.00	14.75 ^b^ ± 0.83	35.16 ^d^ ± 2.04	46.04 ^d^ ± 3.12	26.28 ^c^ ± 2.14	31.81 ^d^ ± 2.20	0 ± 0.00	19.19 ^b^ ± 0.88	0 ± 0.00	0 ± 0.00
**Astragalus**	**Lc** **40 μL/mL**	**Lc** **80** **μL/mL**	**Lg** **40** **μL/mL**	**Lg** **80** **μL/mL**	**Lpc** **40** **μL/mL**	**Lpc** **80** **μL/mL**	**Lpl** **40 μL/mL**	**Lpl** **80** **μL/mL**	**Lrh 40 μL/mL**	**Lrh 80** **μL/mL**
*Acinetobacter* *baumannii*	0 ± 0.00	0 ± 0.00	0 ± 0.00	0 ± 0.00	0 ± 0.00	0 ± 0.00	0 ± 0.00	5.02 ^a^ ± 0.44	0 ± 0.00	0 ± 0.00
*Escherichia* *coli*	0 ± 0.00	0 ± 0.00	10.85 ^b^ ± 0.85	0 ± 0.00	0 ± 0.00	8.17 ^a^ ± 0.42	0 ± 0.00	0 ± 0.00	0 ± 0.00	11.53 ^b^ ± 0.88
*Listeria* *monocytogenes*	0 ± 0.00	47.60 ^d^ ± 2.94	14.21 ^b^ ± 1.33	30.58 ^d^ ± 2.57	0 ± 0.00	0 ± 0.00	34.68 ^d^ ± 2.64	34.97 ^d^ ± 2.88	45.01 ^d^ ± 2.05	48.98 ^d^ ± 2.14
*Pseudomonas* *aeruginosa*	0 ± 0.00	0 ± 0.00	0 ± 0.00	0 ± 0.00	0 ± 0.00	0 ± 0.00	0 ± 0.00	0 ± 0.00	0 ± 0.00	0 ± 0.00
*Staphylococcus* *aureus*	0 ± 0.00	7.06 ^a^ ± 0.84	33.22 ^d^ ± 1.57	50.52 ^d^ ± 2.25	0 ± 0.00	4.06 ^a^ ± 0.34	10.86 ^b^ ± 0.78	46.53 ^d^ ± 1.44	1.11 ± 0.57	45.13 ^d^ ± 2.35
**Carob**	**Lc** **40 μL/mL**	**Lc** **80** **μL/mL**	**Lg** **40** **μL/mL**	**Lg** **80** **μL/mL**	**Lpc** **40** **μL/mL**	**Lpc** **80** **μL/mL**	**Lpl** **40 μL/mL**	**Lpl** **80** **μL/mL**	**Lrh 40 μL/mL**	**Lrh 80** **μL/mL**
*Acinetobacter* *baumannii*	0 ± 0.00	0 ± 0.00	0 ± 0.00	33.05 ^d^ ± 2.25	0 ± 0.00	0 ± 0.00	0 ± 0.00	0 ± 0.00	0 ± 0.00	0 ± 0.00
*Escherichia* *coli*	0 ± 0.00	0 ± 0.00	14.78 ^b^ ± 1.11	36.97 ^d^ ± 2.76	7.92 ^a^ ± 0.37	32.13 ^d^ ± 2.24	0 ± 0.00	38.58 ± 2.44	0 ± 0.00	14.79 ^b^ ± 0.77
*Listeria* *monocytogenes*	0 ± 0.00	31.48 ^d^ ± 2.21	2.42 ^a^ ± 0.2	43.65 ^d^ ± 3.86	3.33 ^a^ ± 0.15	12.16 ^b^ ± 0.88	13.44 ^b^ ± 0.42	42.67 ^d^ ± 3.34	16.81 ^b^ ± 1.44	39.12 ^d^ ± 2.24
*Pseudomonas* *aeruginosa*	0 ± 0.00	0 ± 0.00	0 ± 0.00	0 ± 0.00	0 ± 0.00	0 ± 0.00	0 ± 0.00	0 ± 0.00	0 ± 0.00	0 ± 0.00
*Staphylococcus* *aureus*	0 ± 0.00	27.32 ^c^ ± 1.67	0 ± 0.00	40.33 ^d^ ± 1.67	0 ± 0.00	4.09 ^a^ ± 0.34	12.50 ^b^ ± 1.05	46.52 ^d^ ± 2.34	0 ± 0.00	0 ± 0.00
**Indigo**	**Lc** **40 μL/mL**	**Lc** **80** **μL/mL**	**Lg** **40** **μL/mL**	**Lg** **80** **μL/mL**	**Lpc** **40** **μL/mL**	**Lpc** **80** **μL/mL**	**Lpl** **40 μL/mL**	**Lpl** **80** **μL/mL**	**Lrh 40 μL/mL**	**Lrh 80** **μL/mL**
*Acinetobacter* *baumannii*	0 ± 0.00	0 ± 0.00	0 ± 0.00	0 ± 0.00	0 ± 0.00	0 ± 0.00	0 ± 0.00	0 ± 0.00	0 ± 0.00	0 ± 0.00
*Escherichia* *coli*	0 ± 0.00	10.61 ^b^ ± 0.45	0 ± 0.00	0 ± 0.00	0 ± 0.00	0 ± 0.00	0 ± 0.00	0 ± 0.00	0 ± 0.00	0 ± 0.00
*Listeria* *monocytogenes*	5.53 ^a^ ± 0.33	36.53 ^d^ ± 2.67	0 ± 0.00	5.38 ^a^ ± 0.34	27.91 ^c^ ± 1.98	0 ± 0.00	0 ± 0.00	0 ± 0.00	6.74 ^a^ ± 0.54	18.61 ^b^ ± 0.88
*Pseudomonas* *aeruginosa*	0 ± 0.00	79.67 ^d^ ± 2.57	0 ± 0.00	0 ± 0.00	0 ± 0.00	0 ± 0.00	0 ± 0.00	0 ± 0.00	0 ± 0.00	0 ± 0.00
*Staphylococcus* *aureus*	0 ± 0.00	13.10 ^b^ ± 0.85	0 ± 0.00	0 ± 0.00	12.84 ^b^ ± 0.48	79.99 ^d^ ± 1.03	8.61 ^a^ ± 0.54	43.12 ^d^ ± 2.24	0 ± 0.00	0 ± 0.00
**Sainfoin**	**Lc** **40 μL/mL**	**Lc** **80** **μL/mL**	**Lg** **40** **μL/mL**	**Lg** **80** **μL/mL**	**Lpc** **40** **μL/mL**	**Lpc** **80** **μL/mL**	**Lpl** **40 μL/mL**	**Lpl** **80** **μL/mL**	**Lrh 40 μL/mL**	**Lrh 80** **μL/mL**
*Acinetobacter* *baumannii*	0 ± 0.00	0 ± 0.00	0 ± 0.00	0 ± 0.00	0 ± 0.00	0 ± 0.00	0 ± 0.00	30.42 ^d^ ± 2.31	0 ± 0.00	36.25 ^d^ ± 2.05
*Escherichia* *coli*	0 ± 0.00	0 ± 0.00	0 ± 0.00	0 ± 0.00	0 ± 0.00	0 ± 0.00	0 ± 0.00	0 ± 0.00	0 ± 0.00	0 ± 0.00
*Listeria* *monocytogenes*	0 ± 0.00	11.46 ^b^ ± 0.82	0 ± 0.00	0 ± 0.00	0 ± 0.00	0 ± 0.00	0 ± 0.00	0 ± 0.00	0 ± 0.00	0 ± 0.00
*Pseudomonas* *aeruginosa*	0 ± 0.00	0 ± 0.00	0 ± 0.00	0 ± 0.00	0 ± 0.00	0 ± 0.00	0 ± 0.00	0 ± 0.00	0 ± 0.00	0 ± 0.00
*Staphylococcus* *aureus*	0 ± 0.00	39.56 ^d^ ± 2.82	0 ± 0.00	0 ± 0.00	0 ± 0.00	0 ± 0.00	0 ± 0.00	18.17 ^b^ ± 0.98	8.22 ^a^ ± 0.34	9.72 ^a^ ± 0.78

Results are the average of three independent experiment ± SD. Lc: *Lacticaseibacillus casei* Shirota; Lg: *Lactobacillus gasseri*; Lpc: *Lacticaseibacillus paracasei* subsp. *Paracasei*; Lpl: *Lactiplantibacillus plantarum*; Lrh: *Lacticaseibacillus rhamnosus*. compared with the positive control. ^a^ p < 0.5; ^b^ p < 0.01; ^c^ p < 0.001; ^d^ p < 0.0001 compared with the average values (ANOVA followed by Dunnett’s multiple comparison test).

## Data Availability

The datasets used and/or analyzed during the current study are available from the corresponding author on reasonable request.

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
