# Peer review of "In Vitro Prebiotic Effects and Antibacterial Activity of Five Leguminous Honeys"

_foods, 2023, doi:10.3390/foods12183338_

Round 1

Reviewer 1 Report

Comments and Suggestions for Authors

In vitro prebiotic effects and antibacterial activity of five leguminous honeys

As much as interesting it is to investigate the prebiotic effects and antibacterial activity of honeys, it has to be emphasized that representative samples should be provided. And representative sampling performed.

However, this was not the case in this study.

Authors stated that five organic monofloral honeys have been studied (alfalfa, astragalus, carob, indigo and sainfoin). Of each honey, only one sample (in triplicate) was analysed.

It is unacceptable to base the research on the only one sample per each floral origin. Particularly if conclusions in discussion (whatever they may be) suggest certain effects as attributed honeys of whole botanical species.

Furthermore, authors do not provide any proof of their botanical origin? Apart the statement of the producers that the honeys were submitted to the requested analysis. Requested by who? What analysis? Absolutely, not acceptable. There is whole array of the analysis that should have been carried out for the purpose of determination of honey botanical origin (physical, chemical, mellisopalynological, sensory), all very well described by the scientific literature.

Furthermore, is there any proof that the samples analysed, are honeys at all?

It seems that the authors are not familiar with the field of honey authenticity and how it should be tackled.

Furthermore, authors also revealed the name of the producers' companies. It raises the ethical concerns stating the names of the producers. This looks more like an advertisement or promotion of certain product or producer, which is definitely not suitable for publication in a scientific journal. Especially if attributing the positive health effects to the honeys of particular owners of companies.

Nonetheless the obvious effort invested in the laboratory work for the assessment of the antibacterial activity of the matrix used, the genuinity and authenticity of the samples have not been verified.

In conclusion, taken into the consideration previously described lack of proper scientific reliability of this study.

Comments on the Quality of English Language

No particular comments

Author Response

As much as interesting it is to investigate the prebiotic effects and antibacterial activity of honeys, it has to be emphasized that representative samples should be provided. And representative sampling performed.

However, this was not the case in this study.

Authors stated that five organic monofloral honeys have been studied (alfalfa, astragalus, carob, indigo and sainfoin). Of each honey, only one sample (in triplicate) was analysed.

It is unacceptable to base the research on the only one sample per each floral origin. Particularly if conclusions in discussion (whatever they may be) suggest certain effects as attributed honeys of whole botanical species. in this case it was very difficult to find more companies that sold this type of product. It could have been easier just for some of the honeys, such as alfalfa, but we preferred to limit ourselves to one company per type of honey, also to stay as much as possible in a territorial area of ​​central-southern Italy. The only honey that we did not find for sale by any company in central-southern Italy was indigo honey, sold by a company in northern Italy. A similar work was performed using different Citrus honey (Scientific Reports 2023/1/19), and other monofloral honey (Microorganisms, 2021).

Furthermore, authors do not provide any proof of their botanical origin? Apart the statement of the producers that the honeys were submitted to the requested analysis. Requested by who? What analysis? Absolutely, not acceptable. There is whole array of the analysis that should have been carried out for the purpose of determination of honey botanical origin (physical, chemical, mellisopalynological, sensory), all very well described by the scientific literature. Thank you for this observation. We didn’t express in a correct sense the sentence. We changed it

Furthermore, is there any proof that the samples analysed, are honeys at all? We don’t understand the meaning of your question. We have bought the products commercialised as honey 100%, and we did not have any doubt that the product, present on the market as honey, couldn’t be it

It seems that the authors are not familiar with the field of honey authenticity and how it should be tackled.

Furthermore, authors also revealed the name of the producers' companies. It raises the ethical concerns stating the names of the producers. This looks more like an advertisement or promotion of certain product or producer, which is definitely not suitable for publication in a scientific journal. Especially if attributing the positive health effects to the honeys of particular owners of companies. In previous papers, when we did not add the name of the companies, the reviewers suggested to add them, and for such reason, we made this also now. After your opinion we avoided to indicated the name, indicating only the Italian region of origin of the products.

Nonetheless the obvious effort invested in the laboratory work for the assessment of the antibacterial activity of the matrix used, the genuinity and authenticity of the samples have not been verified. In Italy, all commercial honeys have to follow the Italian law 179 of 2004. In the case of the monofloral honey, the companies must respect the rules on the floral or vegetable origin, if the product is wholly or mainly obtained from the indicated plant and possesses its organoleptic, physicochemical and microscopic characteristics.

In conclusion, taken into the consideration previously described lack of proper scientific reliability of this study. Obviously we respect your opinion

Reviewer 2 Report

Comments and Suggestions for Authors

Overall, the study investigates the prebiotic effects and antibacterial activity of five leguminous honeys. It examines the impact of these honeys on the growth, adhesion, and antioxidant capacity of probiotic bacteria, as well as their ability to inhibit biofilm formation by pathogenic strains. Here is a critical analysis of the study:

Strengths:

·        The study addresses an important topic by exploring the potential prebiotic and antibacterial properties of leguminous honeys.

·        The inclusion of multiple probiotic strains and pathogenic strains adds depth to the findings.

·        The evaluation of the honey's impact on growth, adhesion, and antioxidant capacity provides comprehensive insights into their effects.

Limitations:

·        The study solely focuses on in vitro effects and does not provide information about the potential physiological impact in a living organism. Further research involving in vivo or clinical studies is necessary to validate the findings.

·        The inhibitory effects of the honeys on biofilm formation by pathogenic strains are mentioned, but specific percentages or detailed analysis of these effects are lacking. Further clarification is needed to understand the extent of the antibacterial activity.

·        The lack of specific data and detailed analysis weakens the conclusion's validity.

·        The authors need to provide clarification regarding the origin of the bacterial strains used in the study.

·        The authors should provide details about the transportation and storage conditions of the honey samples that were analyzed.

Comments on the Quality of English Language

Overall, the study investigates the prebiotic effects and antibacterial activity of five leguminous honeys. It examines the impact of these honeys on the growth, adhesion, and antioxidant capacity of probiotic bacteria, as well as their ability to inhibit biofilm formation by pathogenic strains. Here is a critical analysis of the study:

Strengths:

·        The study addresses an important topic by exploring the potential prebiotic and antibacterial properties of leguminous honeys.

·        The inclusion of multiple probiotic strains and pathogenic strains adds depth to the findings.

·        The evaluation of the honey's impact on growth, adhesion, and antioxidant capacity provides comprehensive insights into their effects.

Limitations:

·        The study solely focuses on in vitro effects and does not provide information about the potential physiological impact in a living organism. Further research involving in vivo or clinical studies is necessary to validate the findings.

·        The inhibitory effects of the honeys on biofilm formation by pathogenic strains are mentioned, but specific percentages or detailed analysis of these effects are lacking. Further clarification is needed to understand the extent of the antibacterial activity.

·        The lack of specific data and detailed analysis weakens the conclusion's validity.

·        The authors need to provide clarification regarding the origin of the bacterial strains used in the study.

·        The authors should provide details about the transportation and storage conditions of the honey samples that were analyzed.

Author Response

  • The study solely focuses on in vitro effects and does not provide information about the potential physiological impact in a living organism. Further research involving in vivo or clinical studies is necessary to validate the findings. Dear Professor, thank you very much for your comment. The second step of our complex work will be to evaluate the impact of these types of honey on some activities, such as the anti-inflammatory, antioxidant, and inhibitory effects against the cholinesterases and tyrosinase. When we obtain the most information, we’ll proceed with in vivo studies with the honey/kinds of honey that will have exhibited the best biological performances.
  • The inhibitory effects of the honeys on biofilm formation by pathogenic strains are mentioned, but specific percentages or detailed analysis of these effects are lacking. Further clarification is needed to understand the extent of the antibacterial activity. The inhibitory effects of the honeys on biofilm formation by the five pathogens used in our experiments is shown in table 5
  • The lack of specific data and detailed analysis weakens the conclusion's validity.
  • The authors need to provide clarification regarding the origin of the bacterial strains used in the study. We provided
  • The authors should provide details about the transportation and storage conditions of the honey samples that were analyzed. The transportation was performed in polystyrene bags. The honey was stored at controlled temperature (20°C), simulating a normal domestic condition.

Reviewer 3 Report

Comments and Suggestions for Authors

In this study, the prebiotic effect and antioxidant capacity of five honey were investigated. The topic is interesting and may provide beneficial information for consumers to choose and eat honey.  

Abstract: the abstract section is very important. There should be compelling data and results. The authors give the best data, but under what conditions? Which of the five kinds of honey is best for improving prebiotic properties? Suggesting rewrite. 

Line 128: The formula is not standard. For example, Microbial adhesion (%) = [(A0-A1)/A0]100. 

Line 162: The formula is also not standard. Please modify.

Lines 223, 228: mesh 0.22 mμ? 80 mλ/ml? What’s mean? There are many similar errors in the manuscript. Please check them carefully.

Lines 271-278: Please added the significance analysis in Figure 1 and 2.

Lines 318-343: In this prat of discussion, the authors argue that the prebiotic effect of honey may come from the compound of plant. However, the authors do not give explicit compounds, suggesting supplementation.

Comments on the Quality of English Language
  • The English language is more readable

Author Response

In this study, the prebiotic effect and antioxidant capacity of five honey were investigated. The topic is interesting and may provide beneficial information for consumers to choose and eat honey.  

Abstract: the abstract section is very important. There should be compelling data and results. The authors give the best data, but under what conditions? Which of the five kinds of honey is best for improving prebiotic properties? Suggesting rewrite. Thank you very much. Your comments helped us to improve the abstract

Line 128: The formula is not standard. For example, Microbial adhesion (%) = [(A0-A1)/A0]100.  Thank you for your comment. We provided to modify accordingly.

Line 162: The formula is also not standard. Please modify. We provided to modify accordingly.

Lines 223, 228: mesh 0.22 mμ? 80 mλ/ml? What’s mean? There are many similar errors in the manuscript. Please check them carefully. Thank you for your right comment. We corrected. Sorry for this error.

Lines 271-278: Please added the significance analysis in Figure 1 and 2. We added it. Now, due to the request of another reviewer, which suggested to add also a picture of the honeys, Figure 1 and 2, in the revised version of the manuscript, begin figure 2 and 3, respectively.

Lines 318-343: In this prat of discussion, the authors argue that the prebiotic effect of honey may come from the compound of plant. However, the authors do not give explicit compounds, suggesting supplementation. As we indicated in the manuscript, in the scientific literature are present papers regarding the composition of raw material, not on the honey.

We are proceeding with the chemical composition of these types of the honey and the analysis of some biological properties, such as the antioxidant, anti-inflammatory and inhibitory of cholinesterases and tyrosinase. Studies are also in progress to better define the composition of the supernatants of the probiotic growth, to better define the potential role of honey as source of postbiotics molecules or mixture of molecules.

Reviewer 4 Report

Comments and Suggestions for Authors

The manuscript describes the interesting properties of selected 5 honeys that may be used in medicine in the future, e.g. in increasing the immunity of the elderly and children, and in preventing the development of many diseases caused by bacteria. In the latter application, the ingredients contained in honey can replace antibiotics, to which many bacteria have become resistant.  But a few issues should be completed in it:

1) Photos of the 5 honeys selected for testing should be presented in the EXPERIMENTAL section.

2) Why were these 5 honeys selected for research? What made it so?

3) The article should be supplemented with chemical compositions of the tested honeys.

4) What compounds included in these honeys can play a significant role in inhibiting the development of the pathogenic bacteria studied. What is the mechanism of action of these compounds?

5) What are the main compounds that increase the antioxidant capacity of the tested commercial probiotics?

Author Response

The manuscript describes the interesting properties of selected 5 honeys that may be used in medicine in the future, e.g. in increasing the immunity of the elderly and children, and in preventing the development of many diseases caused by bacteria. In the latter application, the ingredients contained in honey can replace antibiotics, to which many bacteria have become resistant.  But a few issues should be completed in it:

1) Photos of the 5 honeys selected for testing should be presented in the EXPERIMENTAL section. We added the picture indicating it as figure 1. Thus the original figures 1 and 2, in the revised version become 2 and 3, respectively.

2) Why were these 5 honeys selected for research? What made it so? Most studies on monofloral honey from legume flowers refer to well-known honey, such as acacia or clover honey. However, the biodiversity that legumes can offer is enormous. Therefore, the heritage that legumes can represent, also from the point of view of the products deriving from them, such as honey, can further increase the added value even of neglected leguminous species. In Italy, around 100,000 hectares are planted with legumes, producing 190,000 tons (https://blog.wetipico.it/i-legumi-italiani-alla-scoperta-della-loro-importanza/ last access July, 2nd , 2023). In particular, there is an increase in the cultivation of chickpeas, lentils, and peas. Italian production differs in quality and cultural importance. The resumption of legume cultivation, particularly in the central and southern regions, has developed small productions of ancient varieties. Thus, throughout the national territory, species of native legumes previously abandoned are being rediscovered. Therefore, even if the numbers are still small, cultivating these plants is fundamental for the plant biodiversity of the country.

3) The article should be supplemented with chemical compositions of the tested honeys. We are proceeding with chemical composition for a future manuscript.

4) What compounds included in these honeys can play a significant role in inhibiting the development of the pathogenic bacteria studied. What is the mechanism of action of these compounds? Several compounds can be involved in the inhibition of bacterial growth and pathogenicity: polyphenols, volatile compounds, which can act or on the biofilm growth or on metabolism of the bacterial cells. In this first work, we have analysed the whole action of the honey /lab supernatants on biofilm growth and metabolic changes. The next step will be to analyse the secondary composition of these honeys and of the supernatants of the probiotic cultures grown in the presence of the honey. Once the composition will be determined, we will try to evaluate the action of the most abundant molecules on the mechanism of pathogenicity

5) What are the main compounds that increase the antioxidant capacity of the tested commercial probiotics? At this step of work we don’t know what compounds can affect the antioxidant capacity of the tested commercial probiotics. Presumably, it may be the polyphenols and antioxidants found in honey. As the above also indicated, for the antimicrobial activity of the honey, future work will be addressed to study the correlation between the most abundant classes of molecules present in the honey /probiotic culture supernatant and the antioxidant activity of the probiotics used in this first work.

Round 2

Reviewer 1 Report

Comments and Suggestions for Authors

Please do see the reviewer's comments in blue...

As much as interesting it is to investigate the prebiotic effects and antibacterial activity of honeys, it has to be emphasized that representative samples should be provided. And representative sampling performed.

However, this was not the case in this study.

Authors stated that five organic monofloral honeys have been studied (alfalfa, astragalus, carob, indigo and sainfoin). Of each honey, only one sample (in triplicate) was analysed.

It is unacceptable to base the research on the only one sample per each floral origin. Particularly if conclusions in discussion (whatever they may be) suggest certain effects as attributed honeys of whole botanical species. in this case it was very difficult to find more companies that sold this type of product. It could have been easier just for some of the honeys, such as alfalfa, but we preferred to limit ourselves to one company per type of honey, also to stay as much as possible in a territorial area of ​​central-southern Italy. The only honey that we did not find for sale by any company in central-southern Italy was indigo honey, sold by a company in northern Italy. A similar work was performed using different Citrus honey (Scientific Reports 2023/1/19), and other monofloral honey (Microorganisms, 2021).

Reviewer position:

If it is difficoult to find more companies that sold this type of product, it would be proper approach of the researchers to deal with proper honey sampling in situ. The remark stays, it is not appropriate (and acceptible) to attribute any effect to one honey type if there is only one sample per type provided! Not to mention the deficiences noted in providing the genuinity of the samples.

Furthermore, authors do not provide any proof of their botanical origin? Apart the statement of the producers that the honeys were submitted to the requested analysis. Requested by who? What analysis? Absolutely, not acceptable. There is whole array of the analysis that should have been carried out for the purpose of determination of honey botanical origin (physical, chemical, mellisopalynological, sensory), all very well described by the scientific literature. Thank you for this observation. We didn’t express in a correct sense the sentence. We changed it

Reviewer position (2):

It is not visible in the text. Authors statement that „The monofloral character was indicated by the companies and respected the Italian law 179 of 2004, which also legislates on the floral or vegetable origin, if the product is wholly or mainly obtained from the indicated plant and possesses its organoleptic, physicochemical and microscopic characteristics“ does not say nothing about it. As remarked in the first review, the statement of the producers on the market is not a scientific proof. Sorry, but this is not acceptable from the empirical point of view.

One more observation. The photo of the honeys you provided shows that the colour, at least some of them, does not match the uniflorality. It furthermore shows that some additional uniflorality assuring should have been done.

Furthermore, is there any proof that the samples analysed, are honeys at all? We don’t understand the meaning of your question. We have bought the products commercialized as honey 100%, and we did not have any doubt that the product, present on the market as honey, couldn’t be it

Reviewer position (2):

This is exactly the problem! If you bought honeys and you believe that these are genuine samples without analyses carried out, that shows the lack of serious empirical approach. Plus, lack of understanding of the present honey market in European Union where even European Commission admits that there is serious problem with adulteration and frauds of honey at the EU market. Precisely because of that, a proper and representative sampling is imperative. If the study is based on the lack of doubt that the samples are real and genuine, well authors should have provided proofs harder than the statement of the producers.

It seems that the authors are not familiar with the field of honey authenticity and how it should be tackled.

Furthermore, authors also revealed the name of the producers' companies. It raises the ethical concerns stating the names of the producers. This looks more like an advertisement or promotion of certain product or producer, which is definitely not suitable for publication in a scientific journal. Especially if attributing the positive health effects to the honeys of particular owners of companies. In previous papers, when we did not add the name of the companies, the reviewers suggested to add them, and for such reason, we made this also now. After your opinion we avoided to indicated the name, indicating only the Italian region of origin of the products.

Reviewer position (2):

Reviewer cannot say nothing about the previous papers where authors tried to present this or similar study. Thank you for accepting the position of the Reviewer.

Nonetheless the obvious effort invested in the laboratory work for the assessment of the antibacterial activity of the matrix used, the genuinity and authenticity of the samples have not been verified. In Italy, all commercial honeys have to follow the Italian law 179 of 2004. In the case of the monofloral honey, the companies must respect the rules on the floral or vegetable origin, if the product is wholly or mainly obtained from the indicated plant and possesses its organoleptic, physicochemical and microscopic characteristics.

Reviewer position (2):

Indeed, they have! However, honeys have to comply to the Council Directive 2001/110/EC of relating to honey, as well. What’s more, the Italian Law 179/2004 is practical transposing of the effects of EU Directive into Italian legislation. However, the fact that companies should follow these rules does necessary means that they are actually doing it. Authors should have been aware of it before they started with the research activities.

And one more thing, you said very well – In Italy, all commercial honeys have to follow the Italian law 179 of 2004, therefore the title of the proposed manuscript should had been differently titled emphasizing that it is about commercially available honeys.

In conclusion, taken into the consideration previously described lack of proper scientific reliability of this study. Obviously we respect your opinion

Reviewer position (2):

Thank you.

Comments on the Quality of English Language

No particular comments...

Author Response

Dear Reviewer,

You said that we shouldn't trust what was declared by the companies that sold the honey and that even, from the photos, it didn't seem to you that they were honey monofloral. I don't know which country you are from, but I can say, being Italian, that in Italy, in addition to a severe law, there is a specialistic police force, the NAS (anti-sophistication units), which carry out checks on the quality and truthfulness of what is stated by any company in the agri-food sector which, if not in compliance, risks closure, definitive or partial, or in any case a large fine, with enormous negative publicity and economic damage.

Reviewer 4 Report

Comments and Suggestions for Authors

The authors did not determine the compounds (and did not propose possible compounds based on literature reports) that may be responsible for the antioxidant activity of the tested honeys, nor did they determine the main compounds that may increase the antioxidant capacity of the tested commercial probiotics. This information is crucial to the description of the conducted research. The text of the manuscript does not describe the mechanism of action of these antibacterial compounds contained in honeys.

Author Response

Dear Professor, thank you very much for your comments. 

As we replied in the first step, we are conducting analyzes aimed at determining the honey composition, and we could have edited the discussion as required